# Top-Ambiguity Samples Matter: Understanding Why Deep Ensemble Works in Selective Classification

**Qiang Ding**[1,2], **Yixuan Cao**[1,2], **Ping Luo**[1,2,3]

[1]Key Lab of Intelligent Information Processing of Chinese Academy of Sciences (CAS),
Institute of Computing Technology, CAS, Beijing 100190, China
[2]University of Chinese Academy of Sciences, Beijing 100049, China
[3]Peng Cheng Laboratory, Shenzhen 518066, China
`{dingqiang22z, caoyixuan, luop}@ict.ac.cn`

## Abstract

Selective classification allows a machine learning model to reject some hard inputs and thus improve the reliability of its predictions. In this area, the ensemble method is powerful in practice, but there has been no solid analysis on why the ensemble method works. Inspired by an interesting empirical result that the improvement of the ensemble largely comes from *top-ambiguity* samples where its member models diverge, we prove that, based on some assumptions, the ensemble has a lower selective risk than the member model for any coverage within a range. The proof is nontrivial since the selective risk is a non-convex function of the model prediction. The assumptions and the theoretical results are supported by systematic experiments on both computer vision and natural language processing tasks.

## 1 Introduction

Although recent years have witnessed the broad applications of deep learning models, their reliability has not been guaranteed, which gives rise to the study of selective classification. For any given deep learning classifier, there might be inputs that the model is not able to classify correctly in practical applications. One approach to overcoming this problem is accurately delimiting the deep learning classifier's application scope so that the classifier rejects to predict over those hard inputs.

To this end, selective classification aims to learn a pair of models consisting of a standard classifier and a *selective function* that decides whether the system should reject the input. Since the classifier is well studied, the study of selective classification focuses on the design of the selective function. A selective function is usually a confidence estimator with a threshold [1], rejecting the input if and only if the estimated confidence is below the threshold. There are four categories of confidence estimation methods: post-hoc methods [2, 3], Bayesian methods [4, 5, 6], learning-based methods [7, 8, 9, 10], and ensemble methods [11]. Details about these methods is provided in Section 3.

Previous work has shown that Deep Ensemble has a good performance in selective classification [11][1], but no solid analysis explains why the ensemble method works. This paper first proposes a theoretical foundation of Deep Ensemble, i.e., with some assumptions, the ensemble has a lower selective risk than the member model for any target coverage within a range. The proof is nontrivial since the selective risk (with the 0/1 loss) are non-convex. We then verify our assumptions and

---

[1]We conducted more comprehensive experiments, showing that Deep Ensemble is still one of the SOTA in Appendix C.

theoretical result on both computer vision and natural language processing tasks. In summary, the contributions of this paper are summarized as follows.

- We show an interesting empirical result that the performance improvement of Deep Ensemble in terms of selective classification dominantly comes from top-ambiguity samples.
- We are the first to prove that the ensemble has a lower selective risk than the member model for any target coverage within a range based on several reasonable assumptions. Our assumptions and analysis results are verified by systematic experiments on the tasks of image classification and text classification.

## 2 Preliminary

A selective classifier comprises a standard classifier and a selective function. Considering a standard classification problem, $\mathbb{X}$ is a feature space, $\mathbb{Y} = \{1, 2, ..., K\}$ is a finite label set, and a classifier is a function the maps $\mathbb{X}$ to $\mathbb{Y}$. A labeled dataset $\mathbb{D} = \{(x_i, y_i)\}_{i=1}^{N} \subseteq \mathbb{X} \times \mathbb{Y}$ is independently sampled from an identical distribution $\mathcal{D}$. Our goal is to learn a *selective classifier* $(f, g)$, where $f$ is a standard classifier, and $g : \mathbb{X} \to \{0, 1\}$ is a selective function that estimates the correctness of $f$'s prediction. Given input $x$, the output of selective classifier $(f, g)$ is

$$(f, g)(x) = \begin{cases} f(x), & \text{if } g(x) = 1 \\ \text{reject}, & \text{if } g(x) = 0 \end{cases}. \tag{1}$$

The goal of selective classifiers is to minimize the *selective risk* subject to a given target *coverage*. The *coverage* of $(f, g)$ is defined to be the probability of $(f, g)$ not rejecting the input, i.e., $\phi(f, g) := \mathbb{E}_{(x,y)\sim\mathcal{D}}[g(x)]$. The *selective risk* of $(f, g)$ is the risk of $f$ conditioned on $g(x) = 1$, i.e., $R(f, g) := \mathbb{E}_{(x,y)\sim\mathcal{D}}[\ell(f(x), y)g(x)]/\mathbb{E}_{(x,y)\sim\mathcal{D}}(g(x))$, where $\ell : \mathbb{Y} \times \mathbb{Y} \to \mathbb{R}$ is a given loss function. Typically, $\ell$ is the 0/1 loss [1, 7, 8, 9]. Based on these definitions, the objective of selective classifiers is formalized as

$$\min_{f, g} R(f, g), \text{s.t. } \phi(f, g) \geq c_{\text{target}},$$

where $c_{\text{target}}$ is the target coverage. Note that the above objective is evaluated only on in-distribution samples, i.e., samples independently drawn the same distribution of the training data, excluding samples with distribution shifts — also known as OOD samples [12].

The selective function is usually realized by a *confidence estimator* $\kappa : \mathbb{X} \to \mathbb{R}$ with a *confidence threshold* $t$: $g(x) = \mathbf{1}_{\kappa(x)\geq t}$, where $\mathbf{1}$ is the indicator function [1]. In this case, we denote the selective classifier as $(f, \kappa; t)$. As the threshold $t$ controls the coverage, a fixed $(f, \kappa)$ can be fitted to different target coverages (with different confidence thresholds). Thus, $(f, \kappa)$'s performance can be evaluated under multiple different coverages, raising metrics such as *risk-coverage curve (RC curve)* [1] and the *Area Under the Risk-Coverage curve (AURC)* [10]. The *RC curve* of $(f, \kappa)$ is a plot of $(f, \kappa)$'s selective risks against its coverages, which depicts the entire performance profile of $(f, \kappa)$. A lower risk-coverage curve suggests a better selective classifier. The *AURC* is a more concise and comprehensive metric. Similar to the RC curve, a lower AURC suggests a better selective classifier.

This paper focuses on analyzing the selective risk of Deep Ensemble [11], which is defined as follows. To obtain a Deep Ensemble $(f_E, \kappa_E)$, $M$ member models are trained in parallel with random initialization of their parameters and random shuffling of the data points. Assume that each member classifier provides a predictive probability distribution $\boldsymbol{\pi}_m = (\pi_m^1, ..., \pi_m^K), 1 \leq m \leq M$ and predicts $f_m(x) = \arg\max_{1 \leq k \leq K} \pi_m^k$. Then Deep Ensemble is a uniformly-weighted soft voting of member models: $\boldsymbol{\pi}_E(x) := \frac{1}{M} \sum_{m=1}^{M} \boldsymbol{\pi}_m(x)$, and its confidence estimator is $\kappa_E(x) = \max_k \pi_E^k(x)$.

## 3 Related Work

**Selective Classification.** A selective function usually relies on a confidence estimator. To realize the confidence estimator, four approaches has been investigated by previous work. **1. Post-hoc methods**, including Softmax Response (SR) [2] and Trust Score [3]. SR leverages the classifier's maximum class probability (MCP) as the confidence score. Trust Score considers both the distance from the sample to the predicted class and the distance from the sample to another nearest class,

using their ratio as the confidence score. **2. Bayesian methods.** MC-Dropout is an easy-to-use Bayesian methods [5], which samples the classifier's parameters through Dropout [13]. Specifically, MC-Droput enables the dropout layer of the classifier at test time, then runs multiple forwards passes, and finally uses the minus variance of the maximum activation of the classifier's softmax layer as the confidence score. A shortcoming of MC-Dropout is its high time cost at test time. **3. Learning-based methods.** There are two categories of learning-based methods: 1. using MCP as the confidence score and only modifying the standard classifier's loss function, including OSP [14], and Reg-curr [10]; 2. modifying both the architecture and the loss function of the classifier, including SelectiveNet [7], Gambler [8], ConfidNet [15], and SAT [9]. SAT is the SOTA selective classifier in terms of non-ensemble methods, which adds an extra class to the classifier as a reject option and proposes a loss function to train the classifier and the confidence estimator simultaneously. **4. Ensemble Methods.** Deep Ensemble is a vanilla classifiers' ensemble combined by soft voting and equipped with an MCP confidence estimator [11], which has achieved the SOTA in uncertainty estimation. Considering the heavy computational overhead of the ensemble, [16, 17] reduced the ensemble's time consumption by ensemble distillation techniques that distill the ensemble's knowledge into an individual model with the ensemble's diversity retained.

**Analyses of Ensemble Methods.** Ensemble methods combine multiple individual models to improve machine learning models' predictive performances (see [18] for a review). The *error-ambiguity decomposition* has been proposed to explain the better performance of the ensemble in regression tasks [19]. However, for classification tasks, there is no such simple and elegant analysis, since the evaluation metrics are non-convex [20]. Thus, the corresponding analysis for classification tasks needs additional assumptions, e.g., unbiased, uncorrelated, and identically distributed estimation errors for the posterior probability distribution [21, 22]. Nevertheless, these assumptions are impractical [22]. In terms of metrics other than 0-1 loss, [23] uses the entropy of the predictive distribution as uncertainty, deriving the decomposition of data uncertainty and knowledge uncertainty of ensembles. Several other works [24, 25, 26] also discuss different forms of ensemble performance decomposition for metrics other than 0-1 loss. Despite these advances in analyzing ensembles, the analysis of the selective risk of the ensemble remains under-explored.

Other related topic of selective classification include **Out-Of-Distribution Detection** (OOD Detection) and **Calibration**, which along with selective classification are subdomains of uncertainty estimation [27]. *OOD detection* (also known as *open set recognition*) aims to detect test samples with semantic distribution shift (e.g., occurrence of new classes) without losing the in-distribution classification accuracy [12]. OOD detection and selective classification share the requirement for in-distribution classification, and their difference lies in their mutually exclusive detection targets: the former detects misclassified in-distribution samples; while the latter detects samples with semantic distribution shifts. *Calibration* tackles the problem of predicting confidence scores that are representative of the true correctness probabilities [28]. Calibration and selective classification focus on different dimensions of uncertain estimation: calibration focuses on the overall confidence level; while selective classification focuses on the relative confidence ranking among the samples [10].

## 4 The Dominant Source of Performance Improvement of Deep Ensemble

Before we analyze the selective risk of Deep Ensemble, we present an experimental result that helps us to focus on the dominant source of performance improvement of Deep Ensemble. The experiment is to examine whether the performance improvement of the ensemble dominantly comes from high-ambiguity samples, where ambiguity is quantified as $S = \sqrt{\frac{\sum_{i=1}^{M} \|\pi_i - \pi_E\|^2}{M-1}}$. We conduct the experiment on datasets CIFAR-10, CIFAR-100 [29], SVHN [30], MRPC [31], MNLI [32], and QNLI [33] (the details of the datasets and the models are provided in Section 6). The procedure of this experiment is as follows. First, we train a member model and an ensemble, and feed them test data. Second, all test samples are sorted by their ambiguities, and we define samples with the top-50% ambiguities as *high-ambiguity samples* and the rest 50% samples as *low-ambiguity samples*[2]. Third, we construct *ensembling on high-ambiguity samples*, i.e., combining the member models (ensembling) on high-ambiguity samples and using a member model on low-ambiguity samples. Mathematically, denoting the median of ambiguities on the dataset as $t$, ensembling on

---

[2]To intuitively see how large are the top-50% ambiguities, we plot the distribution of ambiguities on each dataset in Appendix F.

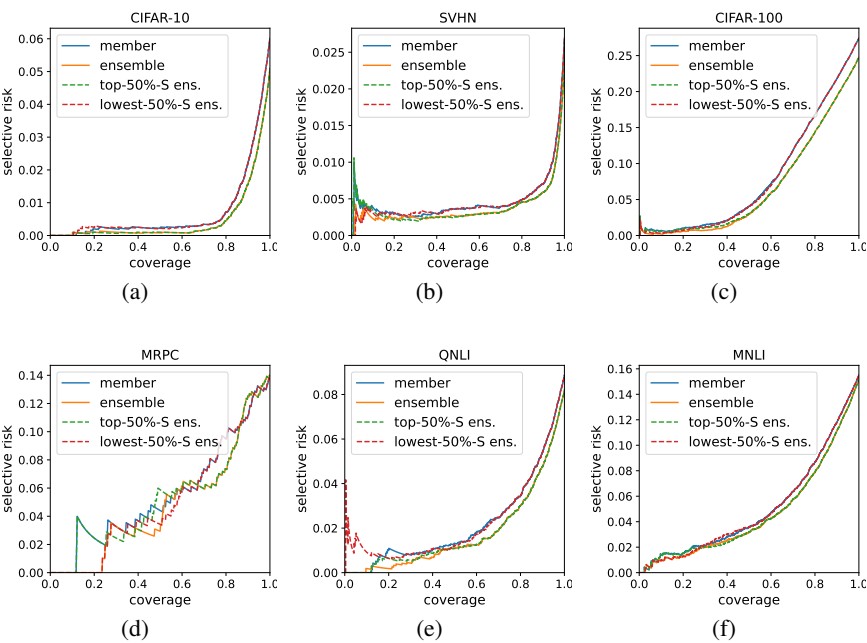

Figure 1: The risk-coverage curves of the member model, Deep Ensemble, ensembling on high-ambiguity samples, and ensembling on low-ambiguity samples on each dataset, where `top/lowest-50%-S ens.` represents ensembling on high/low-ambiguity samples.

high-ambiguity samples induces a model $\tilde{f}_E$ that makes predictions as

$$\tilde{f}_E(x) = \begin{cases} f_E(x), (\text{echoing the prediction of the ensemble}) \text{ if } S(x) \geq t; \\ f_m(x), (\text{echoing the prediction of a member model}), \text{otherwise.} \end{cases}$$

We equate this model with *ensembling on high-ambiguity samples* in the following. Similarly, we define *ensembling on low-ambiguity samples* as combining the member models (ensembling) on low-ambiguity samples and using a member model on high-ambiguity samples. Finally, we plot risk-coverage curves of the member model, the ensemble, ensembling on high-ambiguity samples, and ensembling on low-ambiguity samples.

Figure 1 shows the results of these risk-coverage curves. As we can see, on each dataset, the risk-coverage curve of ensembling on high-ambiguity samples almost coincides with that of the ensemble, and risk-coverage of ensembling on low-ambiguity samples almost coincides with that of the member model. This result suggests that ensembling on high-ambiguity samples provides the dominant performance improvement of the ensemble, while ensembling on low-ambiguity samples provides a minor performance improvement.

## 5 Analysis

This section aims to prove that the selective risk of Deep Ensemble is lower than its member models for any given coverage within a range. If the selective risk is a convex function of member models' predictive probability distributions, then the definition of the ensemble guarantees that the ensemble's selective risk is less than or equal to that of the member model according to Jensen's inequality. However, the selective risk is not a convex function of member models' predictive probability distributions due to the 0/1 loss. Thus, the definition of the ensemble is not sufficient to prove the superiority of the ensemble. The same problem exists in the analysis of error rate in standard classification tasks. To address this problem in classification, three *unrealistic* assumptions: unbiased estimation error, identically distributed estimation error, and uncorrelated estimation error have been proposed [22]. However, we develop other assumptions that are more *realistic* to make our analysis practical.

## 5.1 Intuition

Our first assumption is motivated by the empirical result in Section 4. In the previous experiment, the samples are divided into two categories: low-ambiguity samples and high-ambiguity samples. The result shows that low-ambiguity samples bring little performance improvement, but high-ambiguity samples bring the dominant performance improvement of the ensemble. For the convenience of theoretical analysis, we approximate these two types of samples as follows: 1. *definite samples* on which all member models produce the same predictive probability distribution; 2. *ambiguous samples* on which member models produce statistically-dependent predictive probability distributions (but their joint distribution is unknown).

Based on this assumption, Figure 2 illustrates the intuition of why ensembling works. Given a target coverage, the corresponding confidence thresholds of the member model and the ensemble are shown as the dashed green line and the solid green line, respectively. The solid blue line represents the PDF of the member's (as well as the ensemble's) confidence score over definite samples, and the dashed/solid red line represents the PDF of the member/ensemble model's confidence score over ambiguous samples. Thus, in the upper subfigure, the area under the blue/red line on the right-hand side of the dashed green line is the probability of the member model *accepting* (i.e., not rejecting) the sample given the sample being definite/ambiguous; and a similar result applies to the ensemble in the lower subfigure. In addition, we assume that the member model is not modest over ambiguous samples and over definite samples, i.e., the right-hand ends of the dashed red line and blue line are above the horizontal axis. We can prove that the ensemble prevents high confidence scores (termed as being *modest*) over ambiguous samples, i.e, the right-hand end of the solid red line falls on the horizontal axis. Thus, through ensembling, the PDF over ambiguous samples moves down from the dashed red curve to the solid red curve when the confidence score is near 1 ( as the red arrows show). On the contrary, the PDF over definite samples stays the same. Thus, if the target coverage is small so that the threshold of the member model is near 1, within accepted samples (on the right-hand side of the dashed green line), the proportion of ambiguous samples drops. The threshold of the ensemble model also drops to retain the target coverage, but the net effect of these two changes is a lower proportion of ambiguous samples (or a higher proportion of definite samples) within accepted samples. In addition, we assume that the error rate of classifiers over definite samples is lower than that over ambiguous samples. Thus, the selective risk becomes smaller through ensembling given this coverage.

In the following, we first introduce the assumptions, then we show that the ensemble is modest over ambiguous samples, which leads to an upper bound of ensembles' selective risk, and finally we prove the superiority of the ensemble model.

## 5.2 Assumptions

This section follows the notations of Section 2. Assume that $(f_*, \kappa_*)$ is any member model of $(f_E, \kappa_E)$, where both $\kappa_*$ and $\kappa_E$ are MCP confidence estimators, whose outputs are in $[\frac{1}{K}, 1]$. For the convenience of notation, let $R_*(t)$ and $\phi_*(t)$ denote $(f_*, \kappa_*; t)$'s selective risk and coverage, respectively; and $R_E(t)$ and $\phi_E(t)$ denote $(f_E, \kappa_E; t)$'s selective risk and coverage, respectively, where $t$ is a confidence threshold.

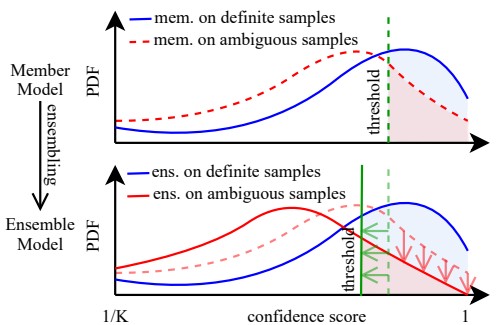

Figure 2: An illustration of the intuition of our analysis. The upper/lower graph shows the PDF of the member/ensemble model's confidence score over ambiguous samples and definite samples.

According to the motivation in Section 4, we assume that the whole data distribution comprises two types of samples: 1. *definite samples* for which all member models provide the same predictive probability distribution; 2. *ambiguous samples* for which member models provide dependent predictive probability distributions (but their dependency is unknown). Formally, we have

**Assumption 1. (Correlation and Diversity Assumption)** Let $a(x) \in \{0, 1\}$ (or $a$ for short) be a hidden variable that denotes whether the sample $x$ is ambiguous ($a = 0/1$ indicates a definite/ambiguous sample). We assume: first, given $a = 0$, $\pi_1 = \pi_2 = \cdots = \pi_M$; second, given

$a = 1, \forall k \in \{1, \ldots, K\}, p(\pi_1^k, \pi_2^k, \ldots, \pi_M^k | a = 1)$ is bounded, i.e., always finite, where $p$ denotes probability density function (PDF).

Assumption 1 depicts both the correlation and the diversity of the ensemble. The existence of definite samples provides a strong correlation among ensemble members' predictions, and the existence of ambiguous samples provides both the correlation and the diversity of the member models' predictions. Furthermore, because Assumption 1 does not specify the correlation of ensemble members over ambiguous samples, this assumption is a relaxation of the uncorrelated-estimation-error assumption of [21, 22].

In addition, because ambiguous samples seem more difficult to classify than definite samples, we assume the member model's selective risk on ambiguous samples is greater than that on definite samples:

**Assumption 2. (Hardness Assumption)** For any member model $(f_*, \kappa_*)$, we assume $\lim_{t \to 1^-} R_*(t|a = 1) > \lim_{t \to 1^-} R_*(t|a = 0)$, where $R_*(t|a)$ is the *conditional selective risk* of $(f_*, \kappa_*; t)$ given $a$, i.e., $\mathbb{E}_{(x,y) \sim \mathcal{D}}[\mathbf{1}_{y \neq f_*(x)} | a, \kappa_*(x) \geq t]$.

Finally, we assume that the member model is *not modest* over some ambiguous samples and some definite samples:

**Assumption 3. (Confidence Assumption)** For any member model $(f_*, \kappa_*)$, we have $\lim_{t \to 1^-} p(\kappa_*(x) = t | a = 0) > 0$ and $\lim_{t \to 1^-} p(\kappa_*(x) = t | a = 1) > 0$.

### 5.3 The Ensemble is Modest over Ambiguous Samples

According to Assumption 3, the member model is *not* modest over ambiguous samples. However, we show that the ensemble is more modest over ambiguous samples (proof is provided in Appendix A):

**Lemma 1.** *Let* $B := \sum_{k=1}^{K} \sup_{[0,1]^M} p(\pi_1^k, \ldots, \pi_M^k | a = 1)$. *Assumption 1 implies* $\forall t \in [0, 1]$, $p(\kappa_E(x) = t | a = 1) \leq M^M B (1 - t)^{M-1}$. *Integrating this inequality, we derive* $P(\kappa_E(x) \geq t | a = 1) \leq M^{M-1} B (1 - t)^M$.

The first inequality in this lemma shows that the ensemble satisfies $\lim_{t \to 1^-} p(\kappa_E(x) = t | a = 1) = 0$, which is the opposite of a member model. In other words, the ensemble is less possible to provide high confidence scores over ambiguous samples than its member models. This result then leads to an upper bound of the ensemble's selective risk.

### 5.4 An Upper Bound of the Ensemble's Selective Risk

We derive an upper bound of the ensemble's selective risk, demonstrating how members' performance and their diversity affect the ensemble's performance. According to selective risk's definition, the ensemble's selective risk can be bounded as (see Appendix A for proof)

$$R_E(t) \leq R_E(t|a = 0) + P(a = 1 | \kappa_E(x) \geq t). \tag{2}$$

The two terms in the right-hand side of (2) can be further reduced to member models' performance and their diversity as follows. First, given $a = 0$, we have $f_E(x) = f_*(x)$ and $\kappa_E(x) = \kappa_*(x)$ (due to Assumption 1), which suggests $R_E(t|a = 0) = R_*(t|a = 0)$. Second, combining the second inequality in Lemma 1 with Bayes' rule, we can bound the second term as follows:

$$P(a = 1 | \kappa_E(x) \geq t) \leq \frac{\gamma \cdot (1 - t)^M}{\gamma \cdot (1 - t)^M + P(\kappa_*(x) \geq t, a = 0)}, \tag{3}$$

where $\gamma = B M^{M-1} \cdot P(a = 1)$. The right-hand side of (3) is negatively related to the diversity of member models. The reason of this statement is as follows. More diverse member models may result in a flatter $p(\pi_1^k, \ldots, \pi_M^k | a = 1)$, leading to a lower $B$. A lower $B$ further leads to a lower right-hand side of (3). Combining the results above, we have

**Lemma 2.** *(Selective Risk Bound of The Ensemble)* *The ensemble's selective risk is bounded as*

$$R_E(t) \leq R_*(t|a = 0) + \frac{\gamma \cdot (1 - t)^M}{\gamma \cdot (1 - t)^M + P(\kappa_*(x) \geq t, a = 0)}. \tag{4}$$

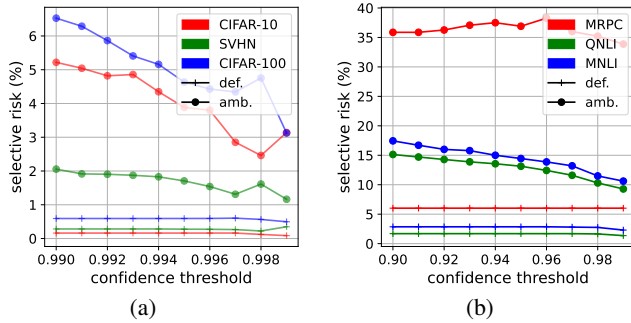

Figure 3: The member model's selective risks on low-ambiguity samples (denoted as `def.`) and on high-ambiguity samples (denoted as `amb.`) given confidence thresholds near 1 on each dataset.

The first term of this upper bound is the selective risk of a member model on definite samples, and the second one is negatively related to the diversity among member models. When member models' performance on definite samples or the diversity improves, this upper bound decreases, which is a reasonable result.

### 5.5 Main Result

Based on Lemma 2, we can prove that Deep Ensemble has a lower selective risk than a member model for any target coverage that is in a range (Theorem 1, see Appendix A for the proof).

**Theorem 1.** *If Assumptions 1, 2, and 3 hold, then $\exists \phi_0 \in (0,1)$ such that $\forall \phi \in (0, \phi_0)$, $\phi_E(t_E) = \phi_*(t_*) = \phi \Rightarrow R_E(t_E) < R_*(t_*)$.*

Furthermore, it is interesting to figure out the maximum $\phi_0$ in Theorem 1. In Appendix E, we use Lemma 2 to estimate the maximum $\phi_0$ (without accessing the ensemble) and find that it can be large, say $> 50\%$, on several datasets. When we instead have the access to the ensemble in experiments, the actual maximums of $\phi_0$ on most datasets are 1, i.e., the largest possible value of $\phi_0$. These results seem interesting, and we will intuitively explain them in the next section.

## 6 Experiments

This section verifies the assumptions and analysis results in realistic tasks, and then compares Deep Ensemble with several SOTA methods and their ensembles in image classification and text classification tasks. However, we do not verify Assumption 1 in the following because the empirical result in Section 4 has shown that Assumption 1 is a reasonable approximation that does not compromise the performance of the ensemble.

**Datasets.** Following [9], we used CIFAR-10, CIFAR-100, and SVHN for image classification tasks. Following [10], we used MRPC, MNLI, and QNLI for text classification tasks. More details of the datasets used in our experiments are described in Appendix B.

**Models.** Following [9, 10], we use VGG-16 [34] and BERT-base [35] as the backbones of selective classifiers for image classification and text classification, respectively. Each member model is built on basis of a backbone model, and an ensemble consists of five member models by default in experiments. More details of the backbone models and their training procedures are provided in Appendix B.

**Verification of Assumptions 2 and 3.** We use samples with ambiguity lower than $\epsilon(\epsilon > 0)$ (termed as *low-ambiguity samples*) to represent definite samples and use samples with ambiguity greater than or equal to $\epsilon$ (termed as *high-ambiguity samples*) to represent ambiguous samples in experiments as we use definite/ambiguous samples to approximate low/high-ambiguity samples in theory. We choose $\epsilon = 10^{-3}$ for datasets of image classification and $\epsilon = 10^{-2}$ for datasets of text classification. Figure 3 shows a member model's conditional selective risks over low-ambiguity samples and high-ambiguity samples against a range of confidence thresholds near 1. The results show that the member model's

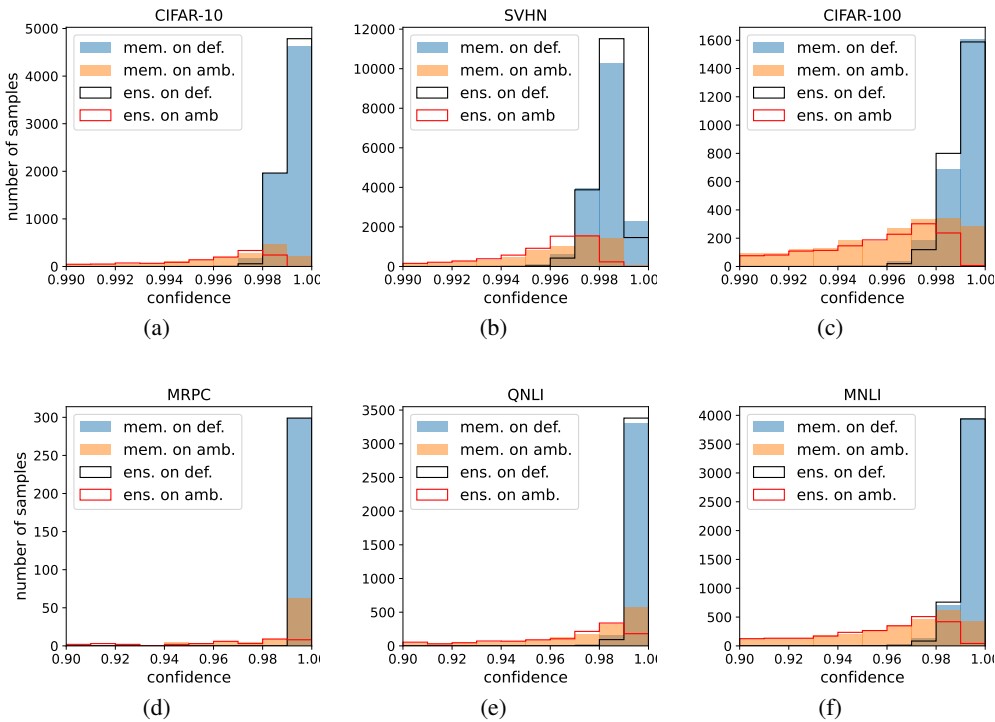

Figure 4: The histograms of the member model's confidence scores and the ensemble's confidence scores on low-ambiguity samples (denoted as `def.`) and on high-ambiguity samples (denoted as `amb.`) for each dataset.

selective risk over low-ambiguity samples is consistently lower than that over high-ambiguity samples given confidence thresholds near 1, which verifies Assumption 2. Figure 4 shows histograms of confidence scores of the member model (as well as the ensemble) on low-ambiguity samples and on high-ambiguity samples. As we can see, the number of low-ambiguity samples and the number of high-ambiguity samples are non-zero in the top bin on each dataset, which verifies Assumption 3. In summary, Assumption 2 and 3 hold on all datasets.

**Verification of Lemma 1.** Figure 4 shows histograms of the ensemble's confidence scores on low-ambiguity samples and on high-ambiguity samples. As we can see, the ensemble's confidence score over high-ambiguity samples has little distribution in the top bin, which is tremendously lower than the corresponding distribution of the member model, while the histogram of confidence score over low-ambiguity samples has almost no change throughout ensembling. This result is consistent with Lemma 1 and our intuition in Figure 2.

**Verification of Theorem 1.** Figure 1 already shows the risk-coverage curves of Deep Ensemble and a member model on each dataset. As we can see, on each dataset except MRPC, the ensemble has a consistently lower selective risk than the member model under any coverage; on MRPC, the ensemble has a lower selective risk than the member model except for coverage of around 30% and around 90%[3]. These results are consistent with Theorem 1.

Despite of this consistency, the theorem does not explain the lower selective risk of the ensemble given a large coverage, say 70%. This is somehow resolved by the analysis in Appendix E. Here, instead of

---

[3]The exception on MRPC might be due to the sparsity of data. The test set of MRPC only contains about 400 examples. Even worse, when the coverage is 30%, the number of accepted examples is much smaller (about 120). These data are inadequate to accurately estimate the selective risk, leading to its high variance. For example, mis-classifying an example by chance could raise the selective risk by about one percent. This high variance might explain why the risk-coverage curve of the member model shakes violently and goes below that of the ensemble several times when the coverage is low. This problem seems irresolvable since we cannot sample more MRPC data to reduce this variance.

diving into theory, we intuitively explain the experimental results as follows. As Lemma 1 claims, $P$(the ensemble yielding confidence $\geq t$|an ambiguous input example) is $O(1 - t)^M (t \to 1^-)$, where $M$ is the number of ensemble members. Therefore, the ensemble hardly provides an ambiguous sample with confidence close to 1. On the contrast, definite samples are always assigned confidence scores close to 1, as the the experimental results show in Figure 4. By this mean, the ensemble stratifies the definite samples and ambiguous sample by their confidence scores, where the definite samples reside on a thin higher layer of confidence than the ambiguous samples (see the rightmost black-edged bars vs. red-edged bars in Figure 4). Combined with the low risk (of both the member model and the ensemble) on definite samples, this stratification lowers the selective risk when the coverage is around the proportion of definite samples in the dataset. Considering the large amount of definite samples (see the heights of black-edged bars in Figure 4), the ensemble model will exhibit lower selective risk than the member model given a considerably large coverage.

In the explanation above, the key factor that leads to the lower selective risk of the ensemble given a large coverage is the experimental fact that the definite samples are large-amounted and always assigned confidence close to 1. This fact is not involved as an assumption in the theory since it seems a very strong assumption (though it holds throughout our experiments). We guess this is attributed to the low bias of DNNs (from a bias-variance perspective), which might be a widespread property of DNNs. Therefore, it would be an interesting direction for future work to strengthen our theory by exploiting this fact.

**Extended Experimental Settings.** To further validate our analysis, we extended the experimental settings on three dimensions: 1. model architecture (to ResNet [36], AlexNet [37], and DenseNet[38]); 2. dataset (to ImageNet100, a subset of ImageNet [39]); 3. number of member models in the ensemble (to 20).

The results are shown in Figure 5. As the figure shows, Assumptions 1, 2, and 3 are consistently verified across various experiment settings. The results indicate that our assumptions might reflect the general characteristics of DNNs. For example, Assumptions 2 and 3 might be attributed to the low bias of DNNs (from a bias-variance perspective) due to DNNs' large model capacity.

These results were unknown when we established our analysis. However, all of them reproduce similar results to the previous results in new settings, further confirming our analysis in practical settings.

# 7 Discussion and Conclusion

We prove that under some assumptions, Deep Ensemble has a lower selective risk than the member model for any target coverage within a range. Although the metrics of selective classification are non-convex, we complete the proof with the help of several assumptions motivated by empirical observations, e.g., the performance improvement of the ensemble dominantly comes from high-ambiguity samples. The assumptions and analysis results are well supported by the experimental results on multiple datasets of image classification tasks and text classification tasks.

Our analysis may benefit the analysis of the Deep Ensemble for standard classification. The previous analyses of the randomization-based ensemble for standard classification require some impractical assumptions [22]. On the contrary, this paper's assumptions are a good approximation of practical settings (see Section 6). Moreover, the standard classification is a subset of selective classification, i.e., selective classification with coverage of 1 (not covered by our analysis result). Thus, our analysis may motivate the analysis of the Deep Ensemble for standard classification in practical settings.

Other possible directions for future work include: 1. extending the $(0, \phi_0)$ mentioned in Theorem 1 to $(0, 1]$, i.e., proving that the ensemble has a lower selective risk than the member model under *any* coverage; 2. adapting the analysis to other selective classifier ensembles, e.g., the ensemble of SAT; 3. relaxing the assumptions to generalize the application scope of our analysis.

# Acknowledgement

This work has been supported by the National Key Research and Development Program of China (No. 2022YFB2702502) and the National Natural Science Foundation of China (No. 62076231,

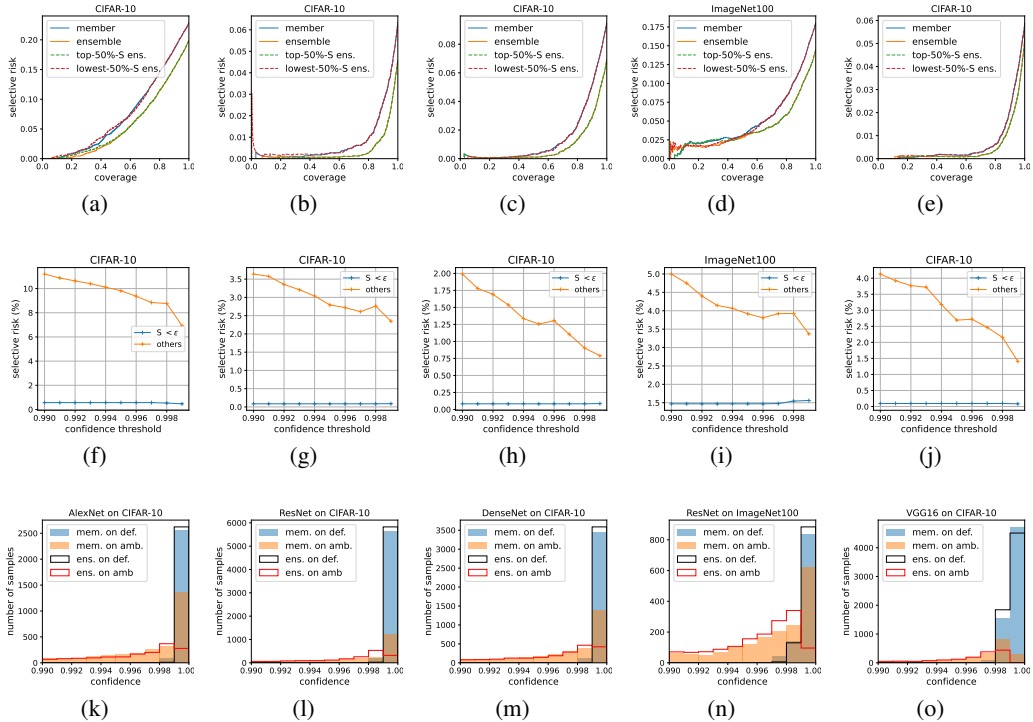

Figure 5: The results of the extended experiments. Here, the 1st/2nd/3rd row reproduces Figure 1/3/4 of the paper in new experiment settings, respectively; each new experiment setting corresponds to a column. Counting from the left, the first three columns show the results of extending model architectures to AlexNet, ResNet50, and DenseNet22 (fixing the dataset to CIFAR10); the 4th column shows the results of extending datasets to ImageNet100 (using the model ResNet50); and the 5th column shows the results of extending the number of member models to 20 (using VGG16 on CIFAR10).

62206265). We thank Keliang Li for his feedback on our theoretical analysis. We also thank anonymous reviewers for their valuable comments.

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

# A Proofs

## A.1 Proof of Lemma 1

The proof of the second inequality in Lemma 1, i.e.,

$$P(\kappa_E(x) \geq t | a = 1) \leq M^{M-1} B(1-t)^M, \tag{5}$$

can be reduced to the proof of the first inequality, i.e.,

$$p(\kappa_E(x) = t | a = 1) \leq M^M B(1-t)^{M-1}. \tag{6}$$

The reason is as follows. If (6) holds, then we have

$$
\begin{aligned}
P(\kappa_E(x) \geq t | a = 1) &= \int_t^1 p(\kappa_E(x) = t' | a = 1) dt' \\
&\leq \int_t^1 M^M B(1-t')^{M-1} \\
&= M^{M-1} B(1-t)^M,
\end{aligned}
$$

which directly derives (5). Therefore, we only need to show (6) at following.

Firstly, we derive the PDF of the average of multiple continuous random variables expressed by the PDFs of these random variables (Lemma 3), which helps us to analyze the PDF of the ensemble's predictive probabilities.

**Lemma 3.** *Let* $X_1, X_2, \ldots, X_M$ *be* $M$ *continuous random variables, and their average is* $X_{\mathrm{avg}} := \frac{1}{M} \sum_{i=1}^M X_i$. *Then the PDF of* $X_{\mathrm{avg}}$[4] *is*

$$p_{X_{\mathrm{avg}}}(x_{\mathrm{avg}}) = M \int_{\mathbb{R}^{M-1}} \mathrm{d}x_1 \mathrm{d}x_2 \cdots \mathrm{d}x_{M-1} \cdot p_{\vec{X}}(x_1, x_2, \ldots, x_{M-1}, Mx_{\mathrm{avg}} - \sum_{i=1}^{M-1} x_i), \tag{7}$$

*where* $p_{\vec{X}}$ *is* $p_{X_1, X_2, \ldots, X_M}$ *for short.*

*Proof.* The distribution function of $X_{\mathrm{avg}}$ is

$$
\begin{aligned}
F_{X_{\mathrm{avg}}}(x_{\mathrm{avg}}) &= \int_{\sum_i x_i \leq Mx_{\mathrm{avg}}} \mathrm{d}x_1 \cdots \mathrm{d}x_{M-1} \mathrm{d}x_M \cdot p_{\vec{X}}(x_1, \ldots, x_M) \\
&= \int_{\mathbb{R}^{M-1}} \mathrm{d}x_1 \cdots \mathrm{d}x_{M-1} \int_{-\infty}^{Mx_{\mathrm{avg}} - \sum_{i=1}^{M-1} x_i} \mathrm{d}x_M \cdot p_{\vec{X}}(x_1, \cdots, x_M).
\end{aligned}
$$

Let $x_M = u - \sum_{i=1}^{M-1} x_i$, then the integral above is equal to

$$
\begin{aligned}
&\int_{\mathbb{R}^{M-1}} \mathrm{d}x_1 \cdots \mathrm{d}x_{M-1} \int_{-\infty}^{Mx_{\mathrm{avg}}} \mathrm{d}u \cdot p_{\vec{X}}(x_1, \ldots, x_{M-1}, u - \sum_{i=1}^{M-1} x_i) \\
&= \int_{-\infty}^{Mx_{\mathrm{avg}}} \mathrm{d}u \int_{\mathbb{R}^{M-1}} \mathrm{d}x_1 \cdots \mathrm{d}x_{M-1} \cdot p_{\vec{X}}(x_1, \ldots, x_{M-1}, u - \sum_{i=1}^{M-1} x_i). \tag{8}
\end{aligned}
$$

The PDF of $X_{\mathrm{avg}}$ is the derivative of $F_{X_{\mathrm{avg}}}$, which, combined with (,8) derives

$$
\begin{aligned}
p_{X_{\mathrm{avg}}}(x_{\mathrm{avg}}) &= F'_{X_{\mathrm{avg}}}(x_{\mathrm{avg}}) \\
&= \frac{\mathrm{d}(Mx_{\mathrm{avg}})}{\mathrm{d}x_{\mathrm{avg}}} \cdot \frac{\mathrm{d}F_{X_{\mathrm{avg}}}}{\mathrm{d}(Mx_{\mathrm{avg}})} \\
&= M \int_{\mathbb{R}^{M-1}} \mathrm{d}x_1 \cdots \mathrm{d}x_{M-1} \cdot p_{\vec{X}}(x_1, \ldots, x_{M-1}, Mx_{\mathrm{avg}} - \sum_{i=1}^{M-1} x_i),
\end{aligned}
$$

which is exactly (7). □

---

[4]A PDF has a subscript to denote which random variable this PDF belongs to.

Secondly, we show the relationship between the PDF of confidence score and the PDFs of predictive probabilities. Note that the confidence score of an SR model is the maximum predictive probability, so the following lemma bounds the PDF of confidence by PDFs of predictive probabilities.

**Lemma 4.** *Let $\Pi^k$ $(1 \leq k \leq K)$ be $K$ continuous random variables, and $C := \max_k \Pi^k$. Then we have*

$$p_C(\kappa) \leq \sum_{k=1}^{K} p_{\Pi^k}(\kappa). \tag{9}$$

*Proof.* First of all, we prove $\forall \kappa_1, \kappa_2, \kappa_1 < \kappa_2$,

$$F_C(\kappa_2) - F_C(\kappa_1) \leq \sum_{k=1}^{K} F_{\Pi^k}(\kappa_2) - F_{\Pi^k}(\kappa_1) \tag{10}$$

It is easy to see that

$$F_C(\kappa) = F_{\Pi^1, \ldots, \Pi^K}(\kappa, \ldots, \kappa) = \int_{(-\infty, \kappa]^K} d\pi^1 \cdots d\pi^K \, p_{\Pi^1, \ldots, \Pi^K}(\pi^1, \cdots, \pi^K),$$

so the left-hand side of (10) is

$$\int_{(-\infty, \kappa_2]^K} d\pi^1 \cdots d\pi^K \, p_{\Pi^1, \ldots, \Pi^K}(\pi^1, \cdots, \pi^K)$$

$$- \int_{(-\infty, \kappa_1]^K} d\pi^1 \cdots d\pi^K \, p_{\Pi^1, \ldots, \Pi^K}(\pi^1, \cdots, \pi^K)$$

$$= \int_{(-\infty, \kappa_2]^K \setminus (-\infty, \kappa_1]^K} d\pi^1 \cdots d\pi^K \, p_{\Pi^1, \ldots, \Pi^K}(\pi^1, \cdots, \pi^K), \tag{11}$$

where the last equality is due to $(-\infty, \kappa_1] \subset (-\infty, \kappa_2]$, and the right-hand side of (10) is

$$\sum_{k=1}^{K} \int_{[\kappa_1, \kappa_2]} d\pi^k \, p_{\Pi^k}(\pi^k)$$

$$= \sum_{k=1}^{K} \int_{\mathbb{R}^{k-1} \times [\kappa_1, \kappa_2] \times \mathbb{R}^{K-k}} d\pi^1 \cdots d\pi^K \cdot p_{\Pi^1, \ldots, \Pi^K}(\pi^1, \cdots, \pi^K)$$

$$\geq \int_{\bigcup_{k=1}^{K} \mathbb{R}^{k-1} \times [\kappa_1, \kappa_2] \times \mathbb{R}^{K-k}} d\pi^1 \cdots d\pi^K \cdot p_{\Pi^1, \ldots, \Pi^K}(\pi^1, \cdots, \pi^K), \tag{12}$$

where the last inequality is because $\mathbb{R}^{k-1} \times [\kappa_1, \kappa_2] \times \mathbb{R}^{K-k}$ for different $k$, $1 \leq k \leq K$, may have an intersection. To prove (10), we only need to prove that the right-hand side of (11) is less than or equal to the right-hand side of (12), which is equivalent to prove

$$(-\infty, \kappa_2]^K \setminus (-\infty, \kappa_1]^K \subset \bigcup_{k=1}^{K} \mathbb{R}^{k-1} \times [\kappa_1, \kappa_2] \times \mathbb{R}^{K-k}. \tag{13}$$

Now we prove (13). $\forall (\pi^1, \ldots, \pi^K) \in (-\infty, \kappa_2]^K \setminus (-\infty, \kappa_1]^K$, we have

$$\forall k, 1 \leq k \leq K, \pi^k \leq \kappa_2, \tag{14}$$

$$\exists k_0, 1 \leq k_0 \leq K, \pi^{k_0} > \kappa_1, \tag{15}$$

where (15) is because if all $\pi^k$ is less than or equal to $\kappa_1$ instead, then $(\pi^1, \ldots, \pi^K) \in (-\infty, \kappa_1]^K$, which contradicts with $(\pi^1, \ldots, \pi^K) \in (-\infty, \kappa_2]^K \setminus (-\infty, \kappa_1]^K$. Thus, $\pi^{k_0} \in [\kappa_1, \kappa_2]$, so

$$(\pi^1, \ldots, \pi^K) \in \mathbb{R}^{k_0 - 1} \times [\kappa_1, \kappa_2] \times \mathbb{R}^{K - k_0} \subset \bigcup_{k=1}^{K} \mathbb{R}^{k-1} \times [\kappa_1, \kappa_2] \times \mathbb{R}^{K-k},$$

which is precisely (13), and therefore (10) is proved.

With (10) and the definition of derivatives, it is easy to see that $F'_C(\kappa) \leq \sum_{k=1}^{K} F'_{\Pi^k}(\kappa)$, which is equivalent to $p_C(\kappa) \leq \sum_{k=1}^{K} p_{\Pi^k}(\kappa)$. Thus, Lemma 4 is proved. $\quad\square$

Based on the lemmas above, we have the following lemma, which is precisely (6). When (6) is proved, the proof of Lemma 1 completes.

**Lemma 5.** *If Assumption 1 holds, then*

$$p(\kappa_E(x) = t|a = 1) \leq M^M B(1-t)^{M-1}. \tag{16}$$

*Proof.* Applying Lemma 3 to the ensemble, we have

$$p(\pi_E^k = t|a = 1) = M \int_{\mathbb{R}^{M-1}} d\pi_1^k \cdots d\pi_{M-1}^k \cdot p(\pi_1^k, \ldots, \pi_{M-1}^k, \pi_M^k = Mt - \sum_{i=1}^{M-1} \pi_i^k|a = 1), \tag{17}$$

The integrand in the right-hand side of (17) being non-zero requires

$$\begin{cases} 0 \leq \pi_i^k \leq 1, i = 1, 2, \ldots, M-1 \\ 0 \leq Mt - \sum_{i=1}^{M-1} \pi_i^k \leq 1 \end{cases}. \tag{18}$$

The inequalities above imply $Mt \leq 1 + \sum_{i=1}^{M-1} \pi_i^k \leq M - 1 + \pi_i^k, \forall i \in \{1, 2, ..., M-1\}$, which further derives $Mt - M + 1 \leq \pi_i^k \leq 1, \forall i \in \{1, 2, ..., M\}$. Thus, (17) can be rewritten as

$$p(\pi_E^k = t|a = 1) = M \int_{[Mt-M+1,1]^{M-1}} d\pi_1^k \cdots d\pi_{M-1}^k \cdot p(\pi_1^k, \ldots, \pi_{M-1}^k, \pi_M^k = Mt - \sum_{i=1}^{M-1} \pi_i^k|a = 1).$$

Considering that $p(\pi_1^k, \ldots, \pi_{M-1}^k, \pi_M^k|a = 1)$ is bounded as Assumption 1 claims, let $B_k$ be its least upper bound. Then we have

$$p(\pi_E^k = t|a = 1) \leq M \int_{[Mt-M+1,1]^{M-1}} d\pi_1^k \cdots d\pi_{M-1}^k B_k = M^M B_k \cdot (1-t)^{M-1}. \tag{19}$$

This inequality combined with Lemma 4 derives $p(\kappa(x) = t|a = 1) \leq \sum_{k=1}^K p(\pi_E^k = t|a = 1) \leq M^M(1-t)^{M-1} \cdot \sum_{k=1}^K B_k$, which is equivalent to the conclusion of this lemma since $B = \sum_{k=1}^K B_k$. □

## A.2 Proof of Equation (2) of the Main Text (Preparing for Lemma 2)

According to the definition of selective risk, we have

$R(f, \kappa; t)$
$= \mathbb{E}_{(x,y)\sim\mathcal{D}}[\mathbf{1}_{y\neq f(x)}|\kappa(x) \geq t]$
$= P(y \neq f(x)|\kappa(x) \geq t)$
$= P(y \neq f(x), a = 0|\kappa(x) \geq t) + P(y \neq f(x), a = 1|\kappa(x) \geq t)$
$= P(y \neq f(x)|a = 0, \kappa(x) \geq t)P(a = 0|\kappa(x) \geq t) + P(y \neq f(x)|a = 1, \kappa(x) \geq t)P(a = 1|\kappa(x) \geq t).$

According to the definition of conditional selective risk, the equation above can be rewritten as

$$R(f, \kappa; t|a = 0)P(a = 0|\kappa(x) \geq t) + R(f, \kappa; t|a = 1)P(a = 1|\kappa(x) \geq t)$$
$$= R(f, \kappa; t|a = 0)[1 - P(a = 1|\kappa(x)] \geq t) + R(f, \kappa; t|a = 1)P(a = 1|\kappa(x) \geq t)$$
$$= R(f, \kappa; t|a = 0) + P(a = 1|\kappa(x) \geq t) \cdot [R(f, \kappa; t|a = 1) - R(f, \kappa; t|a = 0)]$$
$$= R(f, \kappa; t|a = 0) + \lambda(f, \kappa; t) \cdot P(a = 1|\kappa(x) \geq t), \tag{20}$$

where $\lambda(f, \kappa; t) := R(f, \kappa; t|a = 1) - R(f, \kappa; t|a = 0)$. Applying (20) to the ensemble and considering $\lambda(f_E, \kappa_E; t) \leq 1$, we derive an upper bound of the ensemble's selective risk

$$R_E(t) \leq R_E(t|a = 0) + P(a = 1|\kappa_E(x) \geq t). \tag{21}$$

Thus, Equation (2) of the main text is proved.

### A.3 Proof of Theorem 1

*Proof.* **Step 1.** We prove

$$\lim_{t \to 1^-} R_E(t) < \lim_{t \to 1^-} R_*(t). \tag{22}$$

Firstly, we prove $\lim_{t \to 1^-} R_E(t) \leq \lim_{t \to 1^-} R_*(t|a = 0)$. According to Lemma 2, we have

$$\begin{aligned}
\lim_{t \to 1^-} R_E(t) &\leq \lim_{t \to 1^-} \left[ R_*(t|a = 0) + \frac{\gamma \cdot (1-t)^M}{\gamma \cdot (1-t)^M + P(\kappa_*(x) \geq t, a = 0)} \right] \\
&= \lim_{t \to 1^-} R_*(t|a = 0) + \lim_{t \to 1^-} \frac{\gamma \cdot (1-t)^M}{\gamma \cdot (1-t)^M + P(\kappa_*(x) \geq t, a = 0)} \\
&= \lim_{t \to 1^-} R_*(t|a = 0) + \lim_{t \to 1^-} \frac{-M\gamma(1-t)^{M-1}}{-M\gamma(1-t)^{M-1} - p(\kappa_*(x) = t, a = 0)} \\
&= \lim_{t \to 1^-} R_*(t|a = 0) \tag{23}
\end{aligned}$$

where the second-to-last equality is due to L'Hospital's rule, and the last equality is due to $\lim_{t \to 1^-} M\gamma(1-t)^{M-1} = 0$ and $\lim_{t \to 1^-} p(\kappa_*(x) = t, a = 0) = P(a = 0) \lim_{t \to 1^-} p(\kappa_*(x) = t|a = 0) > 0$ (Assumption 3).

Secondly, we prove $\lim_{t \to 1^-} R_*(t) > \lim_{t \to 1^-} R_*(t|a = 0)$. According to (20), we have

$$\lim_{t \to 1^-} R_*(t) = \lim_{t \to 1^-} R_*(t|a = 0) + \lim_{t \to 1^-} \lambda(f_*, \kappa_*; t) \cdot \lim_{t \to 1^-} P(a = 1|\kappa_*(x) \geq t). \tag{24}$$

Using Bayes' rule and L'Hospital's rule, we have

$$\begin{aligned}
\lim_{t \to 1^-} P(a = 1|\kappa_*(x) \geq t) &= \lim_{t \to 1^-} \frac{P(\kappa_*(x) \geq t|a = 1)P(a = 1)}{P(\kappa_*(x) \geq t|a = 1)P(a = 1) + P(\kappa_*(x) \geq t|a = 0)P(a = 0)} \\
&= \lim_{t \to 1^-} \frac{p(\kappa_*(x) = t|a = 1)P(a = 1)}{p(\kappa_*(x) = t|a = 1)P(a = 1) + p(\kappa_*(x) = t|a = 0)P(a = 0)} \\
&> 0, \tag{25}
\end{aligned}$$

where the last inequality is due to Assumption 3. Applying this inequality along with $\lim_{t \to 1^-} \lambda(f*, \kappa_*; t) = \lim_{t \to 1^-} [R_*(t|a = 1) - R_*(t|a = 1)] > 0$ (Assumption 2) to (24), we have

$$\lim_{t \to 1^-} R_*(t) > \lim_{t \to 1^-} R_*(t|a = 0). \tag{26}$$

Combining (23) and (26), we derive (22).

**Step 2.** We prove the ensemble's confidence threshold and the member model's confidence threshold approach 1 when their coverage approach 0. Due to $\phi_*(t) = P(\kappa_*(x) \geq t) = P(\kappa_*(x) \geq t|a = 0)P(a = 0) + P(\kappa_*(x) \geq t|a = 1)P(a = 1)$, we derive

$$\begin{aligned}
\frac{d\phi_*(t)}{dt} &= -p(\kappa_*(x) = t|a = 0)P(a = 0) - p(\kappa_*(x) = t|a = 1)P(a = 1) \\
&\leq -p(\kappa_*(x) = t|a = 0)P(a = 0). \tag{27}
\end{aligned}$$

Because $\lim_{t \to 1^-} p(\kappa_*(x) = t|a = 0) > 0$ (Assumption 3), there exists $\delta_1 > 0$, such that $p(\kappa_*(x) = t|a = 0) > 0, \forall t \in (1 - \delta_1, 1)$. Combining this with (27), we have $\frac{d\phi_*(t)}{dt} < 0, \forall t \in (1 - \delta_1, 1)$ and thus $\phi_*(t)$ is reversible on $(1 - \delta_1, 1)$. Considering $p(\kappa_E(x) = t|a = 0) = p(\kappa_*(x) = t|a = 0)$, we derive $\phi_E(t)$ is reversible on $(1 - \delta_1, 1)$ with the same reasoning as above.

Let $\phi_*^{-1}$ and $\phi_E^{-1}$ be the reverse functions of $\phi_*$ and $\phi_E$ on $(1 - \delta_1, 1)$, respectively. It is easy to see that $\phi_*^{-1}$ is a continuous function, and $\phi_*^{-1}(0) = 1$. Therefore,

$$\lim_{\phi \to 0^+} \phi_*^{-1}(\phi) = 1. \tag{28}$$

For the ensemble, we similarly have

$$\lim_{\phi \to 0^+} \phi_E^{-1}(\phi) = 1. \tag{29}$$

**Step 3.** Combining (22), (28) and (29), we have $R_E(\phi_E^{-1}(\phi)) < R_*(\phi_*^{-1}(\phi))$ when $\phi \to 0^+$, which is equivalent to the result of Theorem 1. $\qquad\square$

Table 1: Sizes of training sets, development sets, and test sets for each dataset used in experiments

| Datasets | Training Set | Development Set | Test Set | Number of Classes |
|---|---|---|---|---|
| CIFAR-10 | 50.0k | | 10.0k | 10 |
| CIFAR-100 | 50.0k | | 10.0k | 100 |
| SVHN | 73.3k | | 26.0k | 10 |
| MRPC | 3.7k | 0.4k | 1.7k | 2 |
| QNLI | 104.7k | 5.5k | 5.5k | 2 |
| MNLI | 392.7k | 9.8k (m)/ 9.8k(mm) | 9.8k(m)/9.8k(mm) | 3 |

Table 2: The value of $o$ on each dataset

| Dataset | CIFAR-10 | SVHN | CIFAR-100 | MRPC | QNLI | MNLI-(m) |
|---|---|---|---|---|---|---|
| $o$ | 2.20 | 2.60 | 4.60 | 1.80 | 1.60 | 2.80 |

# B  Details of Experiments

## B.1  Datasets

The experiments were conducted on multiple data sets of image classification and text classification. The image classification datasets are CIFAR-10, CIFAR-100, [29] and SVHN [30], whose image sizes are all $32 \times 32 \times 3$ pixels. The datasets of text classification are MRPC [31], MNLI [32] and QNLI [33]. The task of MRPC is to judge whether two paragraphs of text are semantically equivalent. MNLI's task is to judge the inferential relationship between sentences (three categories). The task of QNLI is to determine whether a paragraph has the answer to a given question. The sizes of the training set, development set, and test set of each data set used in experiments are shown in Table 1. MNLI's development set and test set are divided into *matched* and *mismatched* parts. In the table, (m) represents matched, and (mm) represents mismatched. The matched parts are sampled from the same source as the training set, while the mismatched parts are sampled from different sources. Current selective classification only considers test samples from the same distribution as the training set, so only the matched parts are used in experiments. In addition, test sets of MRPC, QNLI, and MNLI are not accessible, so we use their development sets as test sets. According to [8, 9], since CIFAR-10, CIFAR-100 and SVHN originally had no development set, their development sets were 2000 samples randomly divided from corresponding test sets.

## B.2  Model Implementations and Training Procedures

For image classification, the backbone model is VGG-16 [34] with Dropout [13], batch normalization [40]. It is trained in the same way as [9]. The model is optimized using SGD with an initial learning rate of 0.1 (the learning rate decays by half in every 25 epochs), the momentum of 0.9, weight decay of 0.0005, batch size of 128, and a total training epoch of 300. Data preprocessing includes data augmentation (random cropping and flip) and normalization. The implementations of the backbone model and data preprocessing are based on the official open-sourced implementation of SAT to ensure a fair comparison.

For text classification, the backbone model of selective classifiers is BERT-base [35]. Pretrained BERT-base is provided by the Huggingface Transformer Library [41]. It is trained/fine-tuned in the same way as [10], except on dataset MRPC. On QNLI and MNLI, the model is trained/fine-tuned using AdamW [42] for 3 epochs, with a learning rate of $2 \times 10^{-5}$, batch size of 32, and the maximum input sequence length of 128. On MRPC, the model is trained/fine-tuned for 10 epoch, with other settings the same as those on QNLI and MNLI. This unique setting of training epoch is due to the small number of samples in MRPC, which makes the training require more epochs to reach convergence on MRPC.

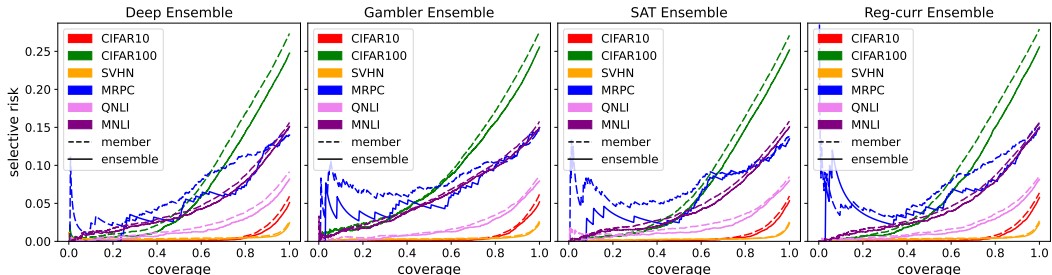

Figure 6: Comparison of the risk-coverage curves between the ensembles (including Deep Ensemble and baselines' ensembles) and their corresponding member models on each dataset.

### B.3 Hyperparameters of Selective Classifiers

For the hyperparameter $o$ of Gambler, we tune $o$ on validation sets in the same way as [8]. The value of $o$ on each dataset is listed in Table 2. For the hyperparameter $\alpha$ of SAT, we set $\alpha = 0.99$, the same as [9]. For the hyperparameter $\lambda$ of Reg-curr, we set $\lambda = 0.05$.

## C  Evaluation of Deep Ensemble

To show that Deep Ensemble is still one of the state-of-the-art, it is necessary to evaluate Deep Ensemble again on selective classification. The reasons are as follows. First, the original paper of Deep Ensemble only reports the accuracy given a confidence threshold, which is not a commonly used metric in selective classification. Second, recent work in selective classification excluded Deep Ensemble as a baseline due to its heavy computational cost [10, 43]. Therefore, no comparison between recent methods and Deep Ensemble are available.

To fill this gap, we report the comparison results of Deep Ensemble and other recent work in this section. We compare Deep Ensemble with three non-ensemble methods: Gambler, SAT, Reg-curr, and their corresponding ensembles. The ensemble method for a baseline is a uniformly-weighted soft voting of the classifiers and the confidence estimators: $\boldsymbol{\pi}_E(x) = \frac{1}{M} \sum_{i=1}^{M} \boldsymbol{\pi}_i(x)$ and $\kappa_E(x) = \frac{1}{M} \sum_{i=1}^{M} \kappa_i(x)$ if the baseline has separate classifier and confidence estimator (i.e., the case of Gambler and SAT); and otherwise, the ensemble method is the same as Deep Ensemble (i.e., the case of Reg-curr). The details of baselines' hyperparameters are provided in Appendix B. Figure 6 shows the risk-coverage curves of Deep Ensemble, baselines, and their corresponding ensembles on each dataset, where each ensemble consists of five member models. The result shows that the ensemble has consistently lower selective risks than the member model under (almost) any coverage across each baseline on each dataset, so we focus on comparing Deep Ensemble with baselines' ensembles. Table 3 shows the AURCs of the ensembles (including Deep Ensemble and baselines' ensembles) and their corresponding member models. Among all ensembles, Deep Ensemble has the lowest AURCs on CIFAR-10, CIFAR-100, and MRPC; SAT ensemble has the lowest AURC on SVHN; and Reg-curr has the lowest AURCs on QNLI and MNLI. Table 4 and 5 shows the selective risks of ensembles under coverage 10%-100% on each dataset. The results show that no ensemble consistently outperforms others under all coverage on all datasets, indicating they have close performances but adopt different trade-offs between coverage and selective risk. In summary, the ensemble largely outperforms more recent methods and is competitive with their ensembles. We hope these results will draw more attention from researchers to the ensemble method.

In addition, we find that the Deep Ensemble can outperform SAT ensemble on SVHN as long as the label noise of SVHN is removed (see Appendix D). These results are surprising considering that Deep Ensemble only uses the MCP as the confidence score, while the baselines adopts more sophisticated techniques to train confidence estimators. Furthermore, we empirically explore further properties of selective classifier ensembles in Appendix D. We show that an ensemble with more member models has a better selective classification performance, and good classification performance does not necessarily imply good selective classification performance.

Table 3: AURC/$10^{-4}$ of the ensembles (including Deep Ensemble and baselines' ensembles) and their corresponding member models on each dataset, where `mem.` and `ens.` represent the member model and the ensemble, respectively. The means and standard deviations are calculated over three trials. The best entries are marked in bold.

| Dataset | | Deep Ensemble | Gambler Ensemble | SAT Ensemble | Reg-curr Ensemble |
|---------|------|---------------|------------------|--------------|-------------------|
| CIFAR-10 | mem. | **66.1±5.6** | 70.7±2.6 | 67.1±0.3 | 67.1±2.0 |
| | ens. | **51.2±2.0** | 57.4±1.0 | 57.6±0.6 | 53.5±0.5 |
| CIFAR-100 | mem. | **793.1±9.0** | 930.6±10.2 | 807.3±6.0 | 851.9±12.8 |
| | ens. | **672.5±1.9** | 857.1±4.7 | 715.5±12.2 | 714.2±6.3 |
| SVHN | mem. | 46.7±0.2 | 44.2±2.5 | **35.0±0.5** | 43.8±1.0 |
| | ens. | 37.1±0.3 | 37.0±0.8 | **32.4±0.9** | 36.6±1.2 |
| MRPC | mem. | **654.3±68.3** | 695.6±65.9 | 794.6±51.9 | 696.8±66.0 |
| | ens. | **561.3±55.6** | 650.1±24.8 | 563.0±16.5 | 640.3±14.1 |
| QNLI | mem. | 221.5 ±8.1 | 235.8 ±35.1 | 246.6 ±4.2 | **201.0 ±1.3** |
| | ens. | 173.3 ±1.8 | 177.7 ±5.3 | 192.9 ±1.8 | **171.9 ±2.4** |
| MNLI | mem. | 515.3 ±8.6 | 607.6 ±19.3 | 519.8 ±8.0 | **496.8 ±9.5** |
| | ens. | 465.2 ±3.8 | 569.4 ±11.4 | 454.4 ±3.1 | **451.9 ±2.1** |

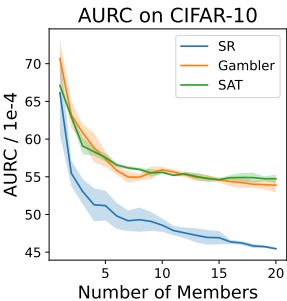

Figure 7: The AURCs on the test set of CIFAR-10 of the SR ensemble (Deep Ensemble), Gambler ensemble, and SAT ensemble of different numbers of members

# D   Further Properties of Selective Classifier Ensemble

## D.1   The Effect of Number of Members on Selective Classifier Ensemble

We evaluate AURCs of the SR ensemble (i.e., Deep Ensemble), Gambler ensemble, and SAT ensemble of different numbers of members on CIFAR10, and find that an ensemble with more members has a better performance, but is less efficient. The results are shown in Figure 7. In most cases, the AURC on the test set of CIFAR-10 decreases as the number of members in the ensemble increases. In addition, as the number of members in the ensemble grows, the effect of adding one member drops. On the one hand, the result shows that an ensemble with a small number of members has good selective classification performance. On the other hand, it indicates that when the number of member models is large, increasing the number of members to improve the performance of the selective classification ensemble is inefficient.

## D.2   Good Classification Performance Does Not Imply Good Selective Classification Performance

It is well known that the ensemble has better classification performance than an individual model, but this does not guarantee a better selective classification performance of the ensemble. To demonstrate

Table 4: The selective risks of ensembles under coverage 10%-100% on image classification datasets. The means and standard deviations are calculated over three trials. The best entries are marked in bold.

| Dataset | coverage (%) | Deep Ensemble | Gambler ensemble | SAT ensemble | Reg-curr ensemble |
|---|---|---|---|---|---|
| CIFAR-10 | 100 | 5.31±0.03 | **5.29±0.03** | 5.47±0.04 | 5.74±0.07 |
|  | 90 | **1.68±0.02** | 1.99±0.01 | 2.15±0.06 | 1.89±0.02 |
|  | 80 | **0.45±0.05** | 0.51±0.02 | 0.63±0.03 | 0.61±0.09 |
|  | 70 | **0.17±0.01** | 0.21±0.01 | 0.26±0.01 | 0.18±0.02 |
|  | 60 | 0.11±0.01 | 0.18±0.03 | 0.17±0.01 | **0.08±0.00** |
|  | 50 | 0.11±0.01 | 0.14±0.02 | 0.11±0.01 | **0.07±0.01** |
|  | 40 | 0.12±0.03 | 0.15±0.02 | **0.06±0.01** | 0.07±0.01 |
|  | 30 | 0.13±0.05 | 0.13±0.03 | **0.06±0.02** | 0.08±0.03 |
|  | 20 | 0.12±0.02 | 0.17±0.05 | **0.00±0.00** | 0.02±0.02 |
|  | 10 | 0.10±0.08 | 0.14±0.05 | **0.00±0.00** | **0.00±0.00** |
| SVHN | 100 | 2.44±0.01 | 2.42±0.02 | **2.36±0.01** | 2.41±0.01 |
|  | 90 | 0.59±0.00 | 0.60±0.03 | **0.50±0.01** | 0.54±0.02 |
|  | 80 | 0.42±0.03 | 0.38±0.01 | **0.34±0.01** | 0.39±0.02 |
|  | 70 | 0.34±0.02 | 0.32±0.01 | **0.31±0.01** | 0.35±0.01 |
|  | 60 | 0.32±0.02 | 0.30±0.01 | **0.28±0.01** | 0.35±0.01 |
|  | 50 | 0.29±0.02 | **0.26±0.01** | **0.26±0.00** | 0.30±0.01 |
|  | 40 | **0.25±0.02** | 0.27±0.02 | **0.25±0.01** | 0.28±0.01 |
|  | 30 | 0.22±0.03 | 0.26±0.01 | **0.20±0.01** | 0.25±0.03 |
|  | 20 | 0.22±0.01 | 0.26±0.01 | **0.18±0.02** | 0.19±0.03 |
|  | 10 | 0.21±0.02 | 0.23±0.00 | **0.17±0.02** | 0.18±0.03 |
| CIFAR-100 | 100 | **24.66±0.08** | 25.50±0.05 | 25.23±0.13 | 25.70±0.09 |
|  | 90 | **19.15±0.15** | 19.88±0.05 | 19.77±0.28 | 20.16±0.14 |
|  | 80 | **14.32±0.22** | 15.75±0.09 | 15.00±0.20 | 15.22±0.07 |
|  | 70 | **9.78±0.13** | 12.11±0.18 | 10.29±0.24 | 10.41±0.38 |
|  | 60 | **5.81±0.06** | 8.89±0.16 | 6.43±0.20 | 6.58±0.27 |
|  | 50 | **2.95±0.04** | 6.22±0.10 | 3.41±0.15 | 3.45±0.05 |
|  | 40 | **1.40±0.13** | 4.37±0.06 | 1.96±0.13 | 1.74±0.11 |
|  | 30 | **0.75±0.05** | 2.67±0.01 | 1.13±0.02 | 0.89±0.06 |
|  | 20 | **0.62±0.06** | 1.91±0.04 | 0.72±0.06 | **0.62±0.04** |
|  | 10 | 0.33±0.09 | 1.42±0.16 | 0.57±0.09 | **0.13±0.05** |

this, we design an SR model with a big backbone, and show that it has as good classification performance as an Deep Ensemble with a standard backbone but worse selective classification performance than an SR model with a standard backbone. The big backbone is designed to have twice as many filters in every convolutional layer and neurons in every fully connected hidden layer as those of the standard VGG-16, which is therefore called *Big VGG-16*. It is easy to see that its number of parameters is approximately $2^2 = 4$ times as many as that of standard VGG-16. We train an Deep Ensemble of 4 VGG-16s and an SR model with a backbone of Big VGG-16 on CIFAR-10 and show the evaluation results in Figure 8 and Table 6. Figure 8 shows that when coverage is high, the ensemble and the big individual model have similar selective risks, and especially, the classification error rates (i.e., selective risk of 100% coverage) of the ensemble and the big individual model are similar. However, when coverage is low, the big individual model has significantly higher selective risk than the ensemble. Table 6 shows that the AURC of Big VGG-16 is much higher than the ensemble of 4 VGG-16s and even higher than SR. In summary, we show that a selective classifier with a good classification performance is not guaranteed to have good selective classification performance, so the good selective classification performance of the ensemble is not a trivial result of its good classification performance.

### D.3 The Effects of Label Noise of SVHN on Selective Classifier Ensembles

In this section, we compare the effect of label noise of SVHN on the Deep Ensemble with that on SAT ensemble, whose result might explain the abnormal experimental results (compared to results on other datasets) on SVHN in Section 6 of the main text. SVHN is not a clean dataset, and much more label noise can be detected in SVHN than in CIFAR-10 and CIFAR-100. Using the soft label of SAT

Table 5: The selective risks of ensembles under coverage 10%-100% on text classification datasets. The means and standard deviations are calculated over three trials. The best entries are marked in bold.

| Dataset | coverage (%) | Deep Ensemble | Gambler ensemble | SAT ensemble | Reg-curr ensemble |
|---------|--------------|---------------|------------------|--------------|-------------------|
| MRPC | 100 | 14.13±0.23 | 14.62±0.23 | **13.64±0.23** | 15.28±0.31 |
|  | 90 | 11.41±1.02 | 11.50±0.46 | **10.69±0.13** | 11.68±0.44 |
|  | 80 | **7.75±0.29** | 9.99±0.88 | 8.46±0.63 | 8.36±0.29 |
|  | 70 | **6.41±0.33** | 8.51±0.72 | 7.93±0.44 | 6.64±0.29 |
|  | 60 | **5.71±0.33** | 7.35±1.20 | 6.39±0.19 | 6.12±0.33 |
|  | 50 | 4.08±1.29 | 4.41±0.40 | **3.59±1.01** | 4.74±0.23 |
|  | 40 | **3.25±0.29** | 3.86±0.76 | **3.25±0.76** | 3.66±0.50 |
|  | 30 | 3.25±0.00 | 3.52±0.38 | 3.52±0.38 | **2.98±0.77** |
|  | 20 | **2.44±1.72** | 3.25±0.58 | 3.25±0.58 | 3.66±0.00 |
|  | 10 | 3.25±2.30 | 4.88±0.00 | **1.63±1.15** | 5.69±1.15 |
| QNLI | 100 | 8.16±0.04 | 8.18±0.20 | **8.03±0.09** | 8.17±0.01 |
|  | 90 | **4.74±0.04** | 5.04±0.14 | 5.03±0.09 | **4.74±0.12** |
|  | 80 | **2.94±0.11** | 3.08±0.08 | 2.97±0.04 | 2.98±0.01 |
|  | 70 | **1.84±0.10** | 1.91±0.04 | 1.92±0.11 | 1.88±0.06 |
|  | 60 | **1.20±0.05** | 1.27±0.05 | 1.36±0.01 | 1.30±0.08 |
|  | 50 | 1.04±0.05 | 1.04±0.08 | 1.13±0.03 | **0.98±0.03** |
|  | 40 | 0.72±0.02 | 0.73±0.04 | 1.02±0.11 | **0.70±0.06** |
|  | 30 | 0.45±0.03 | 0.51±0.03 | 0.83±0.08 | **0.43±0.09** |
|  | 20 | **0.30±0.11** | 0.37±0.15 | 0.67±0.04 | **0.30±0.09** |
|  | 10 | 0.30±0.09 | **0.12±0.09** | 0.61±0.34 | **0.12±0.09** |
| MNLI | 100 | 15.04±0.06 | **14.82±0.15** | 15.03±0.06 | 14.89±0.14 |
|  | 90 | **11.01±0.11** | 11.78±0.09 | 11.37±0.06 | 11.21±0.17 |
|  | 80 | **7.93±0.08** | 9.62±0.08 | 8.26±0.20 | 8.02±0.07 |
|  | 70 | **5.81±0.04** | 8.03±0.23 | **5.81±0.12** | 5.85±0.22 |
|  | 60 | 4.28±0.07 | 6.41±0.25 | 4.08±0.19 | **4.05±0.10** |
|  | 50 | 3.22±0.04 | 4.95±0.10 | **2.99±0.18** | 3.04±0.10 |
|  | 40 | 2.69±0.11 | 3.57±0.21 | **2.08±0.03** | 2.22±0.06 |
|  | 30 | 2.13±0.14 | 2.35±0.18 | **1.57±0.03** | 1.75±0.06 |
|  | 20 | **1.34±0.10** | 1.53±0.17 | 1.39±0.13 | 1.36±0.06 |
|  | 10 | 0.98±0.05 | 1.32±0.14 | **0.71±0.22** | **0.71±0.08** |

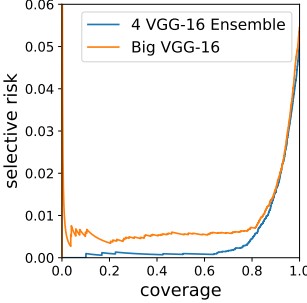

Figure 8: Risk-coverage curves of the ensemble of 4 VGG-16s and the Big VGG-16 on CIFAR-10

[9], we detect label noise in SVHN, CIFAR-10, and CIFAR-100, and find that SVHN has significantly more label noise than CIFAR-10 and CIFAR-100. The result is presented in the following. In addition, it is known that SAT is robust to label noise [9], while SR is not so, so we conjecture that the label noise of SVHN is why the Deep Ensemble is inferior to SAT on SVHN.

We detect label noise with the help of the soft label of SAT. For a sample $x_i$, the soft label of SAT [9], $t_{i,y_i}$, is used to measure $x_i$'s learning difficulty. The soft label of SAT is initialized as 1 and updated

Table 6: The AURCs($/10^{-4}$) of Big VGG-16, a vanilla VGG-16, and the ensemble of 4 VGG-16s on CIFAR-10. The best entries are marked in bold.

| Dataset | Big VGG-16 | VGG-16 | Ensemble |
|---------|-----------|--------|----------|
| CIFAR-10 | 89.2 | 69.6 | **49.3** |

Table 7: Numbers of mislabeled samples in the top-0.1% difficult training samples of SVHN, CIFAR-10, and CIFAR-100.

| Dataset | #Mislabeled | #Top-0.1% | Proportion |
|---------|-------------|-----------|------------|
| SVHN | 73 | 73 | 100% |
| CIFAR-10 | 1 | 50 | 2% |
| CIFAR-100 | 1 | 50 | 2% |

at every training epoch as below

$$t_{i,y_i} \leftarrow \alpha \times t_{i,y_i} + (1 - \alpha) \times p_\theta(y_i|x_i),$$

where $p_\theta(Y|x)$ is the predictive probability distribution of the classifier, $y_i$ is the label of $x_i$, $\alpha$ is a hyperparameter. The smaller the $t_{i,y_i}$ is, the lower the true class predictive probability of the classifier on $x_i$ during training time, indicating that $x_i$ is more difficult to learn. By selecting a percentage of samples with the lowest $t_{i,y_i}$, we get the most difficult samples to learn for the classifier, from which we can easily detect label noise manually.

In training sets of SVHN, CIFAR-10, and CIFAR-100, we detect label noise manually among the top-0.1% difficult (measured by the soft label of SAT) samples. The numbers of mislabeled samples detected in SVHN, CIFAR-10, and CIFAR100 are shown in Table 7. The result shows that SVHN has significantly more mislabeled samples detected than CIFAR-10 and CIFAR-100, indicating much more label noise in SVHN than in CIFAR-10 and CIFAR-100.

To verify the effect of label noise, the following experiments are designed. Firstly, we detect label noise manually among the 1% of the hardest-to-learn samples of SVHN training set and test set, using the soft label of SAT. Secondly, we remove the detected mislabeled samples from the original dataset. The remaining SVHN dataset is called the *clean SVHN*. Accordingly, the original dataset is called the *original SVHN*. Finally, we retrain and test the Deep Ensemble and SAT ensemble and compare their test results. In the second step, the reason for removing mislabeled samples rather than modifying them is that some samples cannot be classified even by humans, and some samples are not in the range of categories of SVHN. Thus, the label noise cannot be eliminated by modifying the labels but by removing mislabeled samples.

The test results of the Deep Ensemble and SAT ensemble on clean SVHN are shown in Table 8. It is not surprising that the AURCs of the Deep Ensemble and SAT ensemble are significantly lower on the clean SVHN than the original SVHN. Furthermore, on the clean SVHN, when the number of members is 5, the AURC of the Deep Ensemble is lower than that of SAT ensemble. Combined with results on the original SVHN, where the AURC of the Deep Ensemble is higher than that of SAT ensemble, we conclude that label noise in SVHN is why the Deep Ensemble has a higher AURC than SAT ensemble. In other words, label noise is why the Deep Ensemble performs worse in selective classification than SAT ensemble on SVHN.

In summary, by experiments, we show that the Deep Ensemble is not as robust to label noise as SAT ensemble, and label noise in SVHN is why the Deep Ensemble is not as good as SAT ensemble on SVHN. We construct the *clean SVHN*, which is SVHN without some mislabeled samples. On the clean SVHN, we compare the Deep Ensemble with SAT ensemble and find that the Deep Ensemble is superior to SAT ensemble in selective classification performance. Combined with former experimental results, we conclude that label noise in SVHN is why the Deep Ensemble is inferior to SAT on SVHN.

Considering the experimental results on the clean SVHN and previous experimental results on CIFAR-10 and CIFAR-100 (see Table 8 and Table 1 of the main text), the Deep Ensemble is superior to SAT

Table 8: AURC/$10^{-4}$ of Deep Ensemble and SAT ensemble on the clean SVHN

| Dataset | #Member | Deep Ensemble | SAT Ensemble |
|---------|---------|---------------|--------------|
| clean SVHN | 1 | 12.3 | **7.8** |
| | 2 | 8.2 | **7.1** |
| | 3 | 7.3 | **6.8** |
| | 4 | **6.8** | **6.8** |
| | 5 | **6.4** | 6.8 |

ensemble in selective classification on clean image classification datasets, Thus, Deep Ensemble is the state-of-the-art selective classification method on clean image classification datasets, but is not as robust to label noise as SAT ensemble.

# E    The Lower Bound of Maximum $\phi_0$ in Theorem 1

This section discusses the lower bound of maximum $\phi_0$ mentioned in Theorem 1. We aim to calculate the maximum $\phi_0$'s lower bound without training an ensemble (otherwise, we can measure it directly on the ensemble).

**Optimization Problem.** To calculate the lower bound of maximum $\phi_0$, we need to solve the following optimization problem

$$\min_{t,t_E} \phi_*(t) \text{ s.t. } \phi_E(t_E) \geq \phi_*(t) \tag{30}$$

$$R_E(t_E) < R_*(t).$$

Using Lemma 2 and $\phi_E(t) \geq P(\kappa_E(x) \geq t, a = 0) = P(\kappa_*(x) \geq t, a = 0)$, we strengthen the constraints of (30) to obtain a looser lower bound of maximum $\phi_0$:

$$\max_{t_*,t_E} \phi_*(t_*) \text{ s.t. } P(\kappa_*(x) \geq t_E, a = 0) \geq \phi_*(t_*)$$

$$R_*(t_E|a = 0) + \frac{\gamma \cdot (1 - t_E)^M}{\gamma \cdot (1 - t_E)^M + P(\kappa_*(x) \geq t_E, a = 0)} < R_*(t_*).$$

For the convenience of solving this optimization problem, we further strengthen the first constraint to obtain a looser lower bound:

$$\max_{t_*,t_E} \phi_*(t_*) \text{ s.t. } P(\kappa_*(x) \geq t_E, a = 0) = \phi_*(t_*) \tag{31}$$

$$R_*(t_E|a = 0) + \frac{\gamma \cdot (1 - t_E)^M}{\gamma \cdot (1 - t_E)^M + P(\kappa_*(x) \geq t_E, a = 0)} < R_*(t_*).$$

**Algorithm.** We design Algorithm 1 to search for the solution to (31), where *oracle* tells whether a sample is definite, and this oracle can be implemented by an ensemble with $M'$ ($M' \ll M$) members. Since $t_E$ is determined by $t_*$ according to the first constraint of (31), (31) can be reduced to a one-dimensional search problem. Our algorithm adopts a binary search for efficiency, although this method might provide a suboptimal solution. Because $\phi_*(t_*)$ is a non-increasing function of $t_*$, Algorithm first search for the minimum $t_*$ by binary search. The procedure of Algorithm 1 in each iteration of the binary search is as follows.

1. Given current $t_*$, Algorithm 1 determines $t_E$ using SEARCHFORTAUENS (see Algorithm 2), a procedure that searches for $t_E \in [0, 1]$ using binary search s.t. $P(\kappa_*(x) \geq t_E, a = 0) = \phi_*(t_*)$. Note that $t_E$ might not exist if $t_*$ is so low that $\phi_*(t_*) > P(a = 0) = \sup_{t_E \in [0,1]} P(\kappa_*(x) \geq t, a = 0)$. This problem will be addressed shortly.

2. Algorithm 1 exams whether $t_E$ exists. If $t_E$ exists, Algorithm 1 then examines whether the second constraint of (31) holds for current $t_*$ and $t_E$, which is implemented by VERIFY-SECONDCONSTRAINT (see Algorithm 3).

**Algorithm 1** A Lower Bound of Maximum $\phi_0$.

---

**Input:** $\kappa_*(\cdot)$, $B$, the test set $\mathcal{D} = \{(x_i, y_i)\}_{i=1}^N$; the oracle $\Omega : \mathbb{X} \to \{0, 1\}$ that tells whether a sample is definite; the number of member models $M$.
**Output:** An lower bound of maximum $\phi_0$ mentioned in Theorem 1
$left = 0$
$right = 1$
$\epsilon = 10^{-9}$
**while** $right - left > \epsilon$ **do**
   $t_* = (left + right)/2$
   $t_E = $ SEARCHFORTAUENS$(t_*, \kappa_*, \mathcal{D}, \Omega)$
   **if** $t_E$ is not None and VERIFYSECONDCONSTRAINT$(\kappa_*, t_*, t_E, \mathcal{D}, \Omega, M, B)$ is True **then**
      $right = t_*$
   **else**
      $left = t_*$
   **end if**
**end while**
$opt = (left + right)/2$
**return** $\frac{1}{N} \sum_{i=1}^N \mathbf{1}_{\kappa_*(x_i) \geq opt}$

---

**Algorithm 2** SEARCHFORTAUENS

---

**Input:** $\kappa_*(\cdot)$, the confidence threshold $t_*$; the test set $\mathcal{D} = \{(x_i, y_i)\}_{i=1}^N$; the oracle $\Omega : \mathbb{X} \to \{0, 1\}$ that tells whether a sample is definite.
**Output:** $t_E \in [0, 1]$ that satisfies the first constraint of (31) given $t_*$.
$\phi = \frac{1}{N} \sum_{i=1}^N \mathbf{1}_{\kappa_*(x_i) \geq t_*}$
**if** $\phi > \frac{1}{N} \sum_{i=1}^N \Omega(x_i)$ **then**
   **return** None
**end if**
$left = 0$
$right = 1$
$\epsilon = 10^{-9}$
**while** $right - left > \epsilon$ **do**
   $t_E = (left + right)/2$
   **if** $\frac{1}{N} \sum_{i=1}^N \mathbf{1}_{\kappa_*(x_i) \geq t_E} \cdot \Omega(x_i) < \phi$ **then**
      $right = t_E$
   **else**
      $left = t_E$
   **end if**
**end while**
**return** $(left + right)/2$

---

3. If $t_E$ exists and the second constraint holds, Algorithm 1 searches for a smaller $t_*$ in the left half feasible area; otherwise, Algorithm 1 searches for a greater $t_*$ in the right half feasible area.

Once the binary search completes, Algorithm 1 returns the coverage of minimum $t_*$.

**Experiment.** To show that Algorithm 1 works in reality, we run this algorithm in the same setting as Section 6 of the main text. In this experiment, $M = 5$, the oracle is implemented by another ensemble with two member models and outputs `True` if and only if the $S < 10^{-3}$. Note that it is difficult to estimate $B$, because: 1. we need to train an ensemble with $M$ models to estimate $B$, which is costly; 2. the domain of $p(\pi_1^k, ..., \pi_M^k | a = 1)$ is of high dimension, so the observed data points are sparse in this domain, which makes the estimation of $B$ more difficult. Thus, we do not estimate $B$ but try several hypothetical values of $B$ to see at what $B$ the lower bound of maximum $\phi_0$ is big.

With different $B$s, we obtain different lower bounds of maximum $\phi_0$ as Table 9 shows. We can see that when the order of magnitude of $B$ is not too big, the lower bound of maximum $\phi_0$ is large and stable, which makes it possible for our algorithm to be used in practical applications. This result also

**Algorithm 3** VERIFYSECONDCONSTRAINT

**Input:** $\kappa_*(\cdot)$; the confidence threshold $t_*$; $t_E$; the test set $\mathcal{D} = \{(x_i, y_i)\}_{i=1}^N$; the oracle $\Omega : \mathbb{X} \to \{0, 1\}$ that tells whether a sample is definite; the number of member models $M$; and $B$.
**Output:** True if and only if $t_*$ and $t_E$ satisfy the second constraint of (31).

$\gamma = BM^{M-1} \frac{1}{N} \sum_{i=1}^N [1 - \Omega(x_i)]$

$leftHandSide = \frac{\sum_{i=1}^N \mathbf{1}_{f(x_i;\theta) \neq y_i} \cdot \Omega(x_i) \cdot \mathbf{1}_{\kappa_*(x_i) \geq t_E}}{\sum_{i=1}^N \Omega(x_i) \cdot \mathbf{1}_{\kappa_*(x_i) \geq t_E}} + \frac{\gamma(1-t_E)^M}{\gamma(1-t_E)^M + \frac{1}{N} \sum_{i=1}^N \Omega(x_i) \cdot \mathbf{1}_{\kappa_*(x_i) \geq t_E}}$

$rightHandSide = \frac{\sum_{i=1}^N \mathbf{1}_{f(x_i;\theta) \neq y_i} \cdot \mathbf{1}_{\kappa_*(x_i) \geq t_*}}{\sum_{i=1}^N \mathbf{1}_{\kappa_*(x_i) \geq t_*}}$

**return** $\mathbf{1}_{leftHandSide < rightHandSide}$

Table 9: The lower bound of maximum $\phi_0$ (%).

| B | 1 | 10 | $10^2$ | $10^3$ | $10^4$ | $10^5$ | $10^6$ | $10^7$ | $10^8$ | $10^9$ | $10^{10}$ | $10^{11}$ | $10^{12}$ | $10^{13}$ | $10^{14}$ |
|---|---|---|---|---|---|---|---|---|---|---|---|---|---|---|---|
| CIFAR-10 | 73.72 | 73.72 | 73.72 | 73.70 | 73.62 | 73.47 | 72.90 | 70.50 | 59.90 | 0 | 0 | 0 | 0 | 0 | 0 |
| SVHN | 77.51 | 77.51 | 77.51 | 77.51 | 77.44 | 77.23 | 76.54 | 68.86 | 0 | 0 | 0 | 0 | 0 | 0 | 0 |
| CIFAR-100 | 32.03 | 32.03 | 32.00 | 32.00 | 31.93 | 31.79 | 31.50 | 30.45 | 26.91 | 17.96 | 10.36 | 0 | 0 | 0 | 0 |
| MNLI | 58.37 | 57.62 | 56.48 | 53.94 | 49.62 | 38.73 | 0 | 0 | 0 | 0 | 0 | 0 | 0 | 0 | 0 |
| QNLI | 65.31 | 65.31 | 65.31 | 64.34 | 62.31 | 59.20 | 53.07 | 37.16 | 0 | 0 | 0 | 0 | 0 | 0 | 0 |
| MRPC | 81.37 | 81.37 | 81.37 | 81.37 | 81.37 | 81.13 | 80.64 | 79.41 | 78.43 | 41.67 | 41.67 | 0 | 0 | 0 | 0 |

indicates the relationship between the ensemble's diversity and its selective classification performance. Since an ensemble with a smaller $B$ seems to have more diversity over ambiguous samples, the result in Table 9 suggests that as long as the ensemble has enough diversity over ambiguous samples, the ensemble is guaranteed to have a lower selective risk than the member model under a considerable range of coverage.

# F   Distributions of Ambiguities

See Figure 9 for the results.

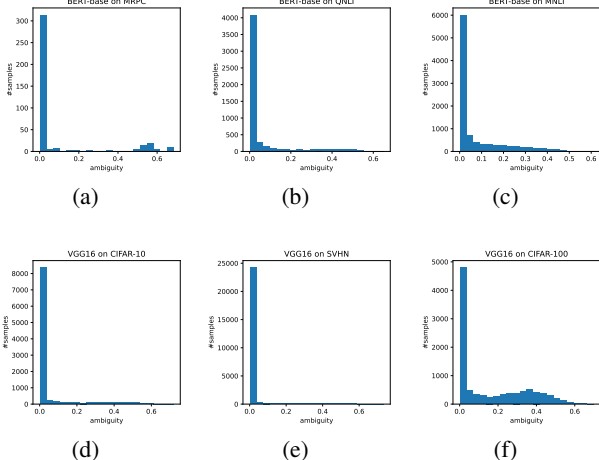

Figure 9: The distribution of ambiguity on different tasks. The pattern of these distributions is clear. For each dataset, the distribution concentrates on a small interval near zero (which we refer to as low-ambiguity samples), and has a long tail (which we refer to as high-ambiguity samples).

