# OpenReview forum: "Top-Ambiguity Samples Matter: Understanding Why Deep Ensemble Works in Selective Classification"
_NeurIPS.cc/2023/Conference — NeurIPS 2023 poster_

### Official Review · Reviewer_qEtZ · 2023-06-29

**Soundness:** 2 fair
**Presentation:** 2 fair
**Contribution:** 3 good
**Rating:** 4
**Confidence:** 4

**Summary:**

This paper focuses on why the ensemble method works.
Authors prove that the ensemble has a lower selective risk than the member model for any coverage within a range, based on some assumptions.
Authors further conduct experiments on both computer vision and natural language processing tasks to verify proofs and assumptions.

**Strengths:**

1. Selective classification in Related Work is summarized well.


**Weaknesses:**

1. The ambiguity in Section 4 is measured based on which model, since there exit an ensemble and a member model. What is the meaning of "ensembling on high-ambiguity samples?" Do you fine-tune models on these "high-ambiguity samples?" If you fine-tune models based on "high-ambiguity samples", why the ambiguity is measured by another ensemble?
Please clarify the above question.

2. Figure 2 is obtained from which experiments or just a sketch? If figure 2 just a sketch, it cannot be used to verify assumption 1.
Authors should conduct massive experiments/proofs to verify this assumption, instead of using a sketch.
Moreover, what is the definition of "correlated predictive probability distributions?" Please clarify it by using mathematic forms.
What is the mathematical definition of "definite samples"? How a sample can be considered as definite?
What is the difference between "definite samples" and "low-ambiguity samples".
Authors use too many different terms without clear definition.

3. Authors should conduct more experiments to verify assumption 1 on more dataset and more DNNs. Just one experiment cannot verify the convincingness of assumption 1.
Moreover, for the verification of Assumptions 2 and 3, authors should use more architectures as the backbone, and construct more ensembles. Just one kind of backbone, and one ensemble cannot sufficiently verify Assumptions 2 and 3.

4. Authors should conduct experiments to explore whether the number of member models affects the performance of the ensemble.


**Questions:**

See weakness.

**Limitations:**

1. Some terms are not defined, and authors do not clarify the difference between different terms, which makes this paper hard to follow.

2.Experimental results are not convincing enough, since authors just use one backbone and one ensemble for each task.

---

> ### Author Rebuttal · Authors · 2023-08-10
>
> Thanks for your valuable comments. This paper is the first to provide the theoretical foundation of Deep Ensemble in selective classification. All the other three reviewers agree that our analysis is sound and insightful. Maybe some points in the paper are not well explained, but we can clarify them now. In addition, we encourage you to read the global response to find out if there are some questions you are interested in. In the following, we will answer your concerns point by point.
>
> > The ambiguity in Section 4 is measured based on which model, since there exist an ensemble and a member model.
>
> The ambiguity is measured based on the divergence of the prediction of multiple-member models. It is defined in Line 109 of the paper.
>
> > What is the meaning of "ensembling on high-ambiguity samples?" Do you fine-tune models on these "high-ambiguity samples?"
>
> As defined in Section 4, *ensembling on high-ambiguity samples* is combining the member models (ensemble) on high-ambiguity samples and using a member model on low-ambiguity samples. By construction, this operation is irrelevant to fine-tuning. Mathematically, this operation introduces a model $\tilde{f}_E$ that makes predictions as
>
> - $\tilde{f}_E(x) = f_E(x)$ (echoing the prediction of the ensemble) if $ambiguity(x)\ge threshold$;
>
> - $\tilde{f}_E(x) = f_m(x)$ (echoing the prediction of a member model), otherwise,
>
> where $threshold$ is the median of ambiguities on the dataset. We equate this model with ensembling on high-ambiguity samples in Section 4.
>
> > Figure 2 is obtained from which experiments or just a sketch? If figure 2 just a sketch, it cannot be used to verify assumption 1. Authors should conduct massive experiments/proofs to verify this assumption, instead of using a sketch.
>
> Figure 2 is a sketch, as its caption indicates ("An illustration of the intuition of our analysis."). This figure is not used to verify Assumption 1, but to illustrate the analysis framework. Assumption 1 is supported by the extensive experiments in Section 4.
>
> > Moreover, what is the definition of "correlated predictive probability distributions?" Please clarify it by using mathematic forms.
>
> Here, we use "correlated" to emphasize that we abandon the *uncorrelated-estimation-error assumption* used in the previous work (on analysis of the ensemble method on ordinary classification tasks, mentioned in Related Work). This assumption states that the errors of member models are uncorrelated. In contrast, we do not specify any form of statistical dependency (including correlation) among member models on ambiguous samples. Therefore, we did not provide a mathematical formula for the definition of "correlated predictive probability distribution".
>
> Considering that these words can be misleading, we are replacing these words with "the statistical dependency among member models is unknown on ambiguous samples".
>
> > What is the mathematical definition of "definite samples"? How a sample can be considered as definite? What is the difference between "definite samples" and "low-ambiguity samples". Authors use too many different terms without clear definition.
>
> We use *low/high-ambiguity samples* in experiment analysis and use *definite/ambiguous samples* in theoretical analysis.
> In experiment analysis, we measure the ambiguity of a sample as the divergency of the prediction among member models. By defining a threshold $\epsilon$, samples with ambiguity less than $\epsilon$ are *low-ambiguity samples*, and others are *high-ambiguity samples*.
> Then, to facilitate the theoretical analysis, we abstract *low-ambiguity samples* as *definite samples* where all member models yield the same predictive probability distribution.
>
> Could you provide some examples of terms that leak definitions to help us improve our paper? The terms questioned in this review are actually defined in the paper:
> - Ambiguity: in line 109,
> - low/high-ambiguity sample: in lines 113-114,
> - ambiguous/definite sample: in lines 145-146, lines 187-189, and in Assumption 1.
>
> > Authors should conduct more experiments to verify assumption 1 on more dataset and more DNNs. Just one experiment cannot verify the convincingness of assumption 1. Moreover, for the verification of Assumptions 2 and 3, authors should use more architectures as the backbone, and construct more ensembles. Just one kind of backbone, and one ensemble cannot sufficiently verify Assumptions 2 and 3.
>
>
> Thank you for your advice. We recently extended our experiments on the following three dimensions:
> 1. the model architectures, extended to ResNet, AlexNet, and DenseNet;
> 2. the datasets, extended to ImageNet100;
> 3. the number of member models, extended to 20.
>
> The results are presented in Figure 0.2 of the global response. As the figure shows, Assumptions 1, 2, and 3 are verified across various experiment settings. The results indicate that our assumptions might reflect the general characteristics of DNNS. This could be explained from a theoretical view. For example, Assumptions 2 and 3 might be attributed to the low bias of DNNs (from a bias-variance perspective) due to DNNs' large model capacity.
>
> > Authors should conduct experiments to explore whether the number of member models affects the performance of the ensemble.
>
> This result is reported in Figure D.1 in the Appendix in the submitted version. The result shows that as the number of member models increases, the AURC of the ensemble decrease. However, the decrease slows down when the number of member models goes up.
>
> If you also care about whether our assumptions are stable across various numbers of member models, you can see the leftmost column of Figure 0.2 in the global response. This figure justifies the assumptions on an ensemble of 20 VGG16 models on CIFAR10.

---

> > ### Comment · Reviewer_qEtZ · 2023-08-17
> >
> > I'm thankful of the authors for their rebuttal response and clarifications.
> > Some of my concerns are addressed.

---

> > > ### Author Response · Authors · 2023-08-17
> > > **Reply to reviewer's comment**
> > >
> > > We are glad to see that some of your concerns are addressed. Will you tell us what concerns remain not resolved?

---

### Official Review · Reviewer_rxSh · 2023-07-05

**Soundness:** 2 fair
**Presentation:** 3 good
**Contribution:** 2 fair
**Rating:** 4
**Confidence:** 3

**Summary:**

This paper aims to investigate why ensemble models perform superiorly compared to member models. The authors conduct an empirical study to demonstrate their assumptions. They separate the data into two categories: high-ambiguity samples and low-ambiguity samples. Their findings reveal that while ensembling high ambiguity samples improve selective risk, the same process for low ambiguity samples yields similar results to those from the member model. Following these observations, they propose several assumptions about the models and samples, demonstrating that under these conditions, ensemble models can achieve a more favorable selective risk. To verify their assumptions and conclusions, the authors conduct additional experiments.

**Strengths:**

1. This paper offers a theoretical understanding of ensemble models, potentially enlightening future algorithm design.
2. Compared to other theoretical frameworks,  the assumptions made in this paper are more realistic.
3. The authors extensively employ empirical experiments to elucidate their motivations and assumptions, enhancing the paper's readability.

**Weaknesses:**

1. Despite the assumption seeming reasonable, the reviewer does not agree that the proof presented in this paper demonstrates the ensemble model's behavior. The proof heavily relies on the $\lim_{t\to 1^-}$ and the main result necessitates a minimal $\phi_0$ value. Nonetheless, as depicted in Figure 5, even with a significantly large $\phi_0$, the ensemble model outperforms the member model. Consequently, the theory proposed in this paper does not align with the results of their experiments.
2. For a theoretical study, the methodologies employed in this paper lack novelty or intrigue.

**Questions:**

What is the intuition behind defining low-ambiguity and high-ambiguity samples as in Assumption 1? Given this assumption, it appears that a majority of samples would be classified as high-ambiguity.

**Limitations:**

The theory provides in this paper can not demonstrate the ensemble model behavior, thus limits the impact of this work.

---

> ### Author Rebuttal · Authors · 2023-08-10
>
> Thanks for your valuable comments. Maybe some points in the paper are not well explained. We will clarify them now. In addition, we encourage you to read the global response where there are some common questions you are interested in. In the following, we will answer your concerns point by point.
>
> > Despite the assumption seeming reasonable, the reviewer does not agree that the proof presented in this paper demonstrates the ensemble model's behavior. The proof heavily relies on the $\lim_{t\rightarrow 1}$ and the main result necessitates a minimal value. Nonetheless, as depicted in Figure 5, even with a significantly large $\phi_0$, the ensemble model outperforms the member model. Consequently, the theory proposed in this paper does not align with the results of their experiments.
>
> We have to point out that the theory **does** align with the results. Theorem 1 states that the ensemble should have a lower selective risk (than the member model) when the target coverage is in $(0, \phi_0) \subset (0, 1]$; and the experiment shows that the ensemble has a lower selective risk when the target coverage is in $(0, 1]$. Our analysis does not state that $\phi_0$ has to be small.
>
> Although our proof relies on $\lim_{t\rightarrow 1}$, in experiments, the confidence score distribution concentrates heavily towards 1. Thus, a $t$ that is close to 1 will still result in a large coverage rate. This explains why in experiments we can observe a high coverage where the ensemble model outperforms the member model. We elaborate on this as follows.
>
>
> As Lemma 1 claims, P(the ensemble yeilding confidence} $\ge t$ | an ambiguous input example) = $O(1-t)^M$ (when $t\rightarrow 1$), where $M$ is the number of ensemble members. Therefore, the ensemble hardly provides an ambiguous sample with confidence close to 1. In contrast, the experiments show that definite samples are always assigned confidence scores that are close to 1 (see the black-edged bars in Figure 4). By this mean, the ensemble stratifies the definite samples and ambiguous samples by their confidence scores, where the definite samples reside on a thin higher layer of confidence than the ambiguous samples (see the rightmost black-edged bars vs. red-edged bars in Figure 4). Combining this stratification with the low risk (of both the member model and the ensemble) on definite samples, when the target coverage is around the proportion of definite samples in the dataset, the selective risk of the ensemble should be lower than that of the member model. Furthermore, considering there are a large number of definite samples (see the heights of black-edged bars in Figure 4), the ensemble model will exhibit lower selective risk than the member model given a considerably large coverage.
>
> In summary, the key factor that leads to the lower selective risk of the ensemble given a large coverage is the experimental fact that the definite samples are large-amounted and always assigned confidence close to 1. This fact is not involved as an assumption in the theory since it seems a very strong assumption (though it holds throughout our experiments). We guess this is attributed to the low bias of DNNs (from a bias-variance perspective), which might be a widespread property of DNNs. Therefore, it would be an interesting direction for future work to strengthen our theory by exploiting this fact.
>
>
> > For a theoretical study, the methodologies employed in this paper lack novelty or intrigue.
>
> This comment seems vague. The contribution of this paper is that it is the first theoretical study on the ensemble model for selective classification. We provide insights into why the ensemble model performs well, propose reasonable assumptions, and give proof. Could you provide a more detailed comment on why our paper lacks novelty? For example, providing existing similar research results.
>
> > What is the intuition behind defining low-ambiguity and high-ambiguity samples as in Assumption 1? Given this assumption, it appears that a majority of samples would be classified as high-ambiguity.
>
> Do you mean the intuition behind definite samples and ambiguous samples defined in Assumption 1? We encourage the reviewer to refer to our general response first. The intuition is that we use *definite samples* (on which all member models yield the same predictive probability distribution) to approximate *low-ambiguity samples* (on which the ambiguity among member models is less than a threshold $\epsilon$). The approximation is equivalent to neglecting ensembling on low-ambiguity samples. This approximation is safe since the experiment in Section 4 shows that ensembling on low-ambiguity samples contributes minor improvement to the performance of the ensemble. In the attached PDF, we also provide the distribution of ambiguity to show that there are many samples on which the member model predictions are quite similar, which serve as low-ambiguity (definite) samples.

---

> > ### Comment · Reviewer_rxSh · 2023-08-15
> > **Thank you for your reply**
> >
> > The reviewer would like to point out that for a theoretical study aiming to explain known phenomena, proper modeling of the real problem is the essential part. The authors argue that their main conclusion is correct, but the correctness of this main conclusion is not what makes their theory to be accepted. After all, the experimental results already demonstrate the effectiveness of the ensemble method. In the authors' response, the authors have pointed out that the confidence distribution in their theory is different from the real distribution, and that's why the reviewer does not give a higher score for the paper: it only analyzes the ensemble model on a small part of the real situation ($\lim_{t\rightarrow 1^-}$). The authors' response does not address this concern, so the reviewer would not increase the score.

---

> > > ### Author Response · Authors · 2023-08-16
> > > **Clarification on reviewer's concern**
> > >
> > > The confidence score distribution concentrates heavily towards 1 in experiments (see Figure 4 in the paper). Although our proof relies on $\lim_{t\rightarrow 1}$, a $t$ close to 1 is able to represent a large proportion of samples so that it can cover the result on a large coverage rate.

---

> > > ### Author Response · Authors · 2023-08-16
> > > **Clarification on reviewer's concern**
> > >
> > > Thanks for your comments. If I understand correctly, your main concern is that the theory cannot explain the lower selective risk of the ensemble given a **large coverage**. We admit that this is not covered by our analysis. However, we can fill the gap by an experiment-based clue. The clue is the stratification of definite samples and ambiguous sapmles, which has been provided in our previous response. If you find it hard to follow in the previous response, we briefly summarize it as follows.
> > >
> > > As Figure 4 shows, in the right-most bar of each subfigure, the ensemble almost clears out ambiguous samples (as Lemma 1 claims) but reserves the definite samples. In addition, the definite samples are dreadfully concentrated to the right-most bar. Thus, ensembling stratifies the definite samples and ambiguous samples and puts the definite samples on a higher level of confidence scores, prioritizing definite samples to select. Combining this with the large amount and low risk of definite samples, the stratification could explain the lower selective risk of the ensemble given a large coverage.
> > >
> > > In summary, the key factor is the distribution of definite samples: a dreadfully concentrated distribution of confidence of definite samples leads to the experimental results. This could not be derived from the theory but we guess this is attributed to the low bias of DNNs (from a bias-variance perspective), which might be a widespread property of DNNs.

---

> > > ### Author Response · Authors · 2023-08-20
> > > **Rebuttal to the reviewer's comment**
> > >
> > > We have to point out that the reviewer misunderstood a key point of our response. The reviewer writes
> > > > In the authors' response, the authors have pointed out that the confidence distribution in their theory is different from the real distribution...
> > >
> > > Actually, the real distribution is a special case of Assumption 3. So, by no means, could one claim that the confidence distribution of our theory is different from the real distribution.
> > >
> > > Opposite to the reviewer's statement, our previous response only demonstrates the specialty of the real distribution (that is not specified in the theory), refilling the gap between the theory and the practice.

---

### Official Review · Reviewer_f6Kz · 2023-07-07

**Soundness:** 4 excellent
**Presentation:** 3 good
**Contribution:** 4 excellent
**Rating:** 8
**Confidence:** 3

**Summary:**

This paper provides a rigorous analysis of the reason for the success of the ensemble method, which includes both empirical evidence and theoretical proof. They found that the power of the ensemble method comes mostly from top-ambiguity samples where the member model diverges, and they provide theoretical evidence as well by proving that the ensemble has a lower selection risk than its member model in certain cases.

**Strengths:**

The paper is well-motivated and well-written.

The experiments are well-designed and thorough.

The theoretical analysis is well-formulated and sound.

The results are insightful and sensible and provide reassuring evidence of the use of ensemble methods in practice.



**Weaknesses:**

The discussion around Figure 2 is in fact a bit hard to digest, the author might want to think about a better way to convey this explanation.



**Questions:**

I don't have any questions.

**Limitations:**

Yes, the author has addressed the limitations.

---

> ### Author Rebuttal · Authors · 2023-08-10
>
> Thanks for your valuable comments. We encourage you to read the global response to find if there are some common questions you are interested in. In the following, We will answer your concern.
>
> > The discussion around Figure 2 is in fact a bit hard to digest, the author might want to think about a better way to convey this explanation.
>
> This seems a challenging task. It would be favorable if you could provide more specific suggestions. Anyway, though, we will try our best to refine our paper.

---

> > ### Comment · Reviewer_f6Kz · 2023-08-16
> >
> > I agree that it is hard to make it even more clear, I am fine with leave it as it is. Overall I have no further questions.

---

### Official Review · Reviewer_Y1pj · 2023-07-07

**Soundness:** 3 good
**Presentation:** 3 good
**Contribution:** 2 fair
**Rating:** 6
**Confidence:** 4

**Summary:**

The authors present an analysis of deep ensembles in the context of selective classification, where a classifier has an option to abstain from providing a response in situations where it lacks confidence in its predictions. They prove that under reasonable assumptions, the performance of a deep ensemble in selective classification is guaranteed to beat that of its component members under the zero-one loss. They go on to justify their assumptions and provide experimental evidence for their claims, and finally demonstrate that deep ensembles are competitive with currently used methods in selective classification.

**Strengths:**

- Compelling analysis. The authors provide an insightful analysis of ensemble performance that operates under reasonable assumptions that they later justify. Their approach offers theoretical insight into an interesting application of ensembles, namely selective classification.
- Related work. To my understanding, the authors provide a thorough and easy to understand survey of other methods in the field of selective classification.

**Weaknesses:**

- I found section 4 of the paper to be weaker than some of the other sections. One instantiation of low vs. high ambiguity samples corresponds to having ensemble members which make identical predictions on 90% of the data, and different predictions only on the remaining 10%, which would also explain the results in Figure 1. While such an effect would still be consistent with the analysis that follows, it would be important to report in its own right. I would like to understand what the distribution of ambiguous samples looks like.
- Although illustrative, I do not follow how the results in Figure 5 and Table 1 correspond to the rest of the paper. In particular, how do these results relate to the importance of ambiguous samples in the performance of deep ensembles?

**Questions:**

- Figure 2 provides a useful intuition for the analysis that follows. Is it possible to provide a version of Figure 2 with real data as a justification of your assumptions as well?
- It may be worthwhile to include references to more recent work in ensemble performance decompositions for metrics besides 0-1 loss:
	- https://arxiv.org/pdf/2206.10566.pdf
	- https://proceedings.mlr.press/v151/ortega22a.html
	- https://arxiv.org/abs/2301.03962,
	- https://openreview.net/forum?id=6sBiAIpkUiO
- The choice of confidence estimator seems reasonable, but I could imagine cases where the maximum confidence does not correspond to the chosen output. Out of interest, could the analysis apply to other confidence estimators?
- In the MRPC dataset, the ensemble has higher selective risk than the member at ~30% coverage, as discussed in the text. Is this a violation of assumptions, or is $\phi_0$ lower than $30%$ for this dataset?
- From your analysis, my understanding is that the ensemble should outperform any individual ensemble member (which is better than the average performance, as guaranteed by Jensen for convex losses). Is Figure 5 showing a comparison to the average single model, or the best, or a randomly selected one?

**Limitations:**

 As discussed in the text, the analysis is limited to selective classification within a range of coverages, and extensions to more general settings are discussed.

---

> ### Author Rebuttal · Authors · 2023-08-10
>
> Thanks for your valuable comments. We encourage you to read the global response to find if there are some common questions you are interested in. In the following, we will answer your concerns point by point.
>
> > I found section 4 of the paper to be weaker than some of the other sections. One instantiation of low vs. high ambiguity samples corresponds to having ensemble members which make identical predictions on 90% of the data, and different predictions only on the remaining 10%, which would also explain the results in Figure 1. While such an effect would still be consistent with the analysis that follows, it would be important to report it in its own right. I would like to understand what the distribution of ambiguous samples looks like.
>
> Excuse me, we find it a little hard to get what your question is. We try to answer the question as follows. The distribution of ambiguities is shown in Figure 0.1 in the global response. As the figure shows, on each dataset, the distribution concentrates on a small interval near 0 as well as exhibits a long tail. The concentration corresponds to the definite samples, and the long tail corresponds to the ambiguous samples. Please comment on this response if you have any further questions.
>
> > Although illustrative, I do not follow how the results in Figure 5 and Table 1 correspond to the rest of the paper. In particular, how do these results relate to the importance of ambiguous samples in the performance of deep ensembles?
>
> Figure 5 can be related to the importance of ambiguous samples as follows:
> 1. the importance of ambiguous samples motivates Assumption 1;
> 2. based on Assumption 1 (as well as the other two assumptions), we prove Theorem 1;
> 3. Theorem 1 is verified by the leftmost subfigure of Figure 5.
>
> The rest parts of Figure 5 and Table 1 are not related to the importance of ambiguous samples, which just compare the performance of the deep ensemble and other recent methods. However, these results could help researchers recognize the powerful performance of deep ensemble more clearly. Since the publication of Deep Ensemble in 2017, more recent work in selective classification did not include Deep Ensemble as a baseline due to its heavy computational cost (Xin et al., 2021; Feng et al., 2023). This narrows the research of selective classification to individual models. To broaden the horizon of selective classification, we report the comparison results of Deep Ensemble and other recent work in Figure 5 and Table 1. As the results show, the ensemble largely outperforms more recent methods and is competitive with their ensembles. We hope these results will draw more attention from researchers to the ensemble method.
>
> Reference
>
> - Xin et al. The Art of Abstention: Selective Prediction and Error Regularization for Natural Language Processing. In ACL, 2021.
> - Feng et al. Towards Better Selective Classification. In ICLR, 2023.
>
> > Figure 2 provides a useful intuition for the analysis that follows. Is it possible to provide a version of Figure 2 with real data as a justification of your assumptions as well?
>
> A version of Figure 2 with real data can be found in Figure 4. Here, each subfigure is an instance of Figure 2 evaluated on a dataset. Although the x-axes are truncated (to focus on the overlapping regions of definite examples and ambiguous examples), and the thresholds are not specified in these subfigures, we can easily find that these subfigures reveal the case of Figure 2.
>
> > It may be worthwhile to include references to more recent work in ensemble performance decompositions for metrics besides 0-1 loss: ...
>
> Thanks for sharing recent papers that analyze ensembles. They extend our knowledge of diversity and performance decompositions of the ensemble. Fortunately, the problems they solved do not coincide with the problem considered in this paper. We would like to include them as related work to our paper.
>
> > The choice of confidence estimator seems reasonable, but I could imagine cases where the maximum confidence does not correspond to the chosen output. Out of interest, could the analysis apply to other confidence estimators?
>
> Theoretically, it could. To apply our analysis to general cases (i.e., the member model is a general selective classifier (f, g), rather than a vanilla classifier in the case of Deep Ensemble), we just need to modify the definition of definite samples and ambiguous samples. The definite samples should be redefined as those examples on which all member models predict the same (f, g) values and the ambiguous samples should be redefined as those examples on which no statistical dependency of member models are specified. As long as we redefine definite/ambiguous samples and claim almost the same assumptions, we can derive the same result as Theorem 1.
>
> Although the extension is simple, in practice, the assumptions should be examined again. This could be an interesting future work.
>
> > In the MRPC dataset, the ensemble has higher selective risk than the member at ~30% coverage, as discussed in the text. Is this a violation of assumptions, or is lower than for this dataset?
>
> The reason might be the sparsity of data. The test set of MRPC only contains about 400 examples. Even worse, when the coverage is 30%, the number of accepted examples is much smaller (about 120). These data are inadequate to accurately estimate the selective risk, leading to its high variance. For example, misclassifying an example by chance could raise the selective risk by about one percent. This high variance might explain why the risk-coverage curve of the member model shakes violently and goes below that of the ensemble several times when the coverage is low. This problem seems irresolvable since we cannot sample more MRPC data to reduce this variance.
>
> > Is Figure 5 showing a comparison to the average single model, or the best, or a randomly selected one?
>
> In Figure 5, each single model is randomly selected from the corresponding ensemble's member models.

---

> > ### Comment · Reviewer_Y1pj · 2023-08-15
> > **Thank you for your response.**
> >
> > - It's very useful to see the distribution of responses, which captures the phenomenon that I mentioned in my initial review (identical predictions on most of the data). Given such a distribution, most "low ambiguity samples" as defined in this case are actually zero-ambiguity samples, so the results in Figure 1 are trivially to be expected. It would be important to include Figure 0.1 in the main text, as it provides a much more definitive view of how ambiguity works in an ensemble than the current Figure 1. In particular, we might imagine a case where ambiguity distributions are much more centrally distributed. Such an ambiguity distribution might still generate risk-coverage curves that look like Figure 1, which would be a much more surprising finding.
> > - Thank you likewise for clarifying the role of the remainder of Figure 5, beyond the leftmost panels.
> > - Thank you for pointing out that Figure 4 demonstrates the intuition given in Figure 2- I missed this in my reading of the text. I'd like to suggest unifying the color scheme/layout between Figure 2 and 4 so this correspondence is easier to see.
> > - I appreciate the answers to my remaining questions as well. I will be keeping my score.

---

### Author Rebuttal · Authors · 2023-08-10

Thank all reviewers for their valuable comments. This paper is the first to provide a theoretical foundation of Deep Ensemble in selective classification. All reviewers, except Reviewer qEtZ, agree that our paper has the following strengths:
1. The analysis in this paper provides some insight into the behavior of the ensemble model on selective classification problems.
2. The assumptions in this paper are more realistic and reasonable than existing works (that analyze ensemble models on ordinary classification tasks). They are well motivated and justified by experiments.

However, it seems that some points on Assumption 1 (about ambiguous and definite samples) are not clear, and the concerns of the reviewers concentrate on Assumption 1. So, we will clarify it here. We also conduct more experiments on more datasets and backbones, as per the Reviewer qEtZ's comments.



**Clarification on Assumption 1.**

We first introduce the intuition behind Assumption 1. Based on the experiment results, we observe that the predictions of member models coincide on some samples and diverge on some other samples. Moreover, the performance improvement of the ensemble model largely comes from the diverged (ambiguous) samples. This motivates us to propose Assumption 1 to facilitate our theoretical analysis.

Concern 1: the definitions and differences of *definite/ambiguous samples* and *low/high-ambiguity samples* (Reviewers rxSh and qEtZ).

We use *low/high-ambiguity samples* in experiment analysis and use *definite/ambiguous samples* in theoretical analysis.
In experiment analysis, we measure the ambiguity of a sample as the divergency of the prediction among member models. By defining a threshold $\epsilon$, samples with ambiguity less than $\epsilon$ are *low-ambiguity samples*, and others are *high-ambiguity samples*.
Then, to facilitate the theoretical analysis, we abstract *low-ambiguity samples* as *definite samples* where all member models yield the same predictive probability distribution, as Assumption 1 states. This is a simplification of the real-world situation.

Concern 2: What is the distribution of ambiguity on a dataset? (Reviewer Y1pj)

We provide the distribution of the ambiguity in Figure 0.1 in the pdf of this response.


**More experimental results.**

We recently extended the experiments on three dimensions:

1. model architecture (to ResNet, AlexNet, and DenseNet);
2. dataset (to ImageNet100);
3. number of member models in the ensemble (to 20).

The results are shown in Figure 0.2 in the pdf of this response. As the figure shows, Assumptions 1, 2, and 3 are consistently verified across various experiment settings. The results indicate that our assumptions might reflect the general characteristics of DNNS. This could be explained from a theoretical view. For example, Assumptions 2 and 3 might be attributed to the low bias of DNNs (from a bias-variance perspective) due to DNNs' large model capacity.

---

### Decision · Program_Chairs · 2023-09-21

**Decision:**

Accept (poster)

**Comment:**

The paper analyzes deep ensembles in the context of selective classification, demonstrating that under specific assumptions, an ensemble's performance is superior to its individual members under zero-one loss. They highlight the superiority of ensembles' performance compared to individual members, specifically when abstaining from low-confidence predictions. To justify their claims, the authors provide a theoretical proof, validated by empirical evidence, where the study hinges on the differentiation between high-ambiguity and low-ambiguity samples to elucidate the performance benefits.

**Strengths:**

- Provides a theoretical understanding of ensemble models, which can influence future algorithm designs.
- The assumptions made are generally perceived as realistic.
- Empirical experiments support the motivations and assumptions, enhancing paper clarity.
- Strong coverage of related work, especially on selective classification.

**Weaknesses:**

- There is ambiguity concerning ensemble members' predictions on the data, and how this connects with Figure 1.
- Both Figure 5 and Table 1 do not seem to clearly align with the paper's broader narrative, particularly regarding ambiguous samples and deep ensembles.
- Some reviewers highlighted discrepancies between the theoretical proof provided and the empirical results presented, indicating potential inconsistencies.
- There's confusion regarding the specific definition of terms like "high-ambiguity samples", "definite samples", and their respective roles within the paper.
- Experimental results were seen as lacking in breadth. The validation of assumptions across datasets and architectures was found to be insufficient. This is partially mitigated by new results in the rebuttal.

The paper seems to present compelling insights into deep ensembles in selective classification. There are inconsistencies however between theoretical and empirical results, coupled with ambiguities in definitions, and a perceived lack of comprehensive evaluation. With that said, new results during the rebuttal seem to more thoroughly validate the assumptions. And I recommend the authors make some of the clarity corrections during the camera-ready.